# COMBO: Compositional World Models for Embodied Multi-Agent Cooperation

**Hongxin Zhang**[1,*] **Zeyuan Wang**[2*], **Qiushi Lyu**[3*], **Zheyuan Zhang**[1], **Sunli Chen**[1],
**Tianmin Shu**[4], **Behzad Dariush**[5], **Kwonjoon Lee**[5], **Yilun Du**[6], **Chuang Gan**[1]
[1] University of Massachusetts Amherst [2] IIIS, Tsinghua University [3] Peking University
[4] Johns Hopkins University [5] Honda Research Institute USA [6] MIT

## Abstract

In this paper, we investigate the problem of embodied multi-agent cooperation, where decentralized agents must cooperate given only egocentric views of the world. To effectively plan in this setting, in contrast to learning world dynamics in a single-agent scenario, we must simulate world dynamics conditioned on an arbitrary number of agents' actions given only partial egocentric visual observations of the world. To address this issue of partial observability, we first train generative models to estimate the overall world state given partial egocentric observations. To enable accurate simulation of multiple sets of actions on this world state, we then propose to learn a compositional world model for multi-agent cooperation by factorizing the naturally composable joint actions of multiple agents and compositionally generating the video conditioned on the world state. By leveraging this compositional world model, in combination with Vision Language Models to infer the actions of other agents, we can use a tree search procedure to integrate these modules and facilitate online cooperative planning. We evaluate our methods on three challenging benchmarks with 2-4 agents. The results show our compositional world model is effective and the framework enables the embodied agents to cooperate efficiently with different agents across various tasks and an arbitrary number of agents, showing the promising future of our proposed methods. More videos can be found at https://umass-embodied-agi.github.io/COMBO/.

## 1 Introduction

Building cooperative embodied agents that can engage in and help humans in tasks requiring visual planning is a valuable yet challenging endeavor. To cooperatively plan in a multi-agent scenario, in contrast to learning world dynamics in a single-agent scenario, there are additional challenges to simulate world dynamics conditioned on joint actions and partial observations of the world.

Large generative models have brought remarkable advances to various domains, including language understanding and generation (OpenAI, 2023), image understanding and generation (Liu et al., 2023a; Ho et al., 2020), and video generation (Blattmann et al., 2023a). Many have explored how to leverage these powerful foundation models for embodied AI, Ahn et al. (2022); Wang et al. (2023a) leverage Large Language Models for decision-making, Zhang et al. (2023) incorporates LLMs to build modular embodied agents for cooperation and communication, Driess et al. (2023) uses Vision Language Models to build capable vision agents, Du et al. (2023b); Yang et al. (2024) use Video Models to generate visual plans for robots and explore modeling world dynamics with video models to improve single-agent planning. However, how to leverage Vision Language Models and Video Models to build embodied agents capable of planning under a visual cooperation task is under-explored where it's important to accurately simulate the world dynamics conditioned on an arbitrary number of agents' actions given only partial egocentric observations at each step for efficient cooperation.

We propose to learn a compositional world model for multi-agent cooperation by leveraging the natural compositionality of joint actions to generate future frames conditioned on the current world state

---

*denotes equal contribution.

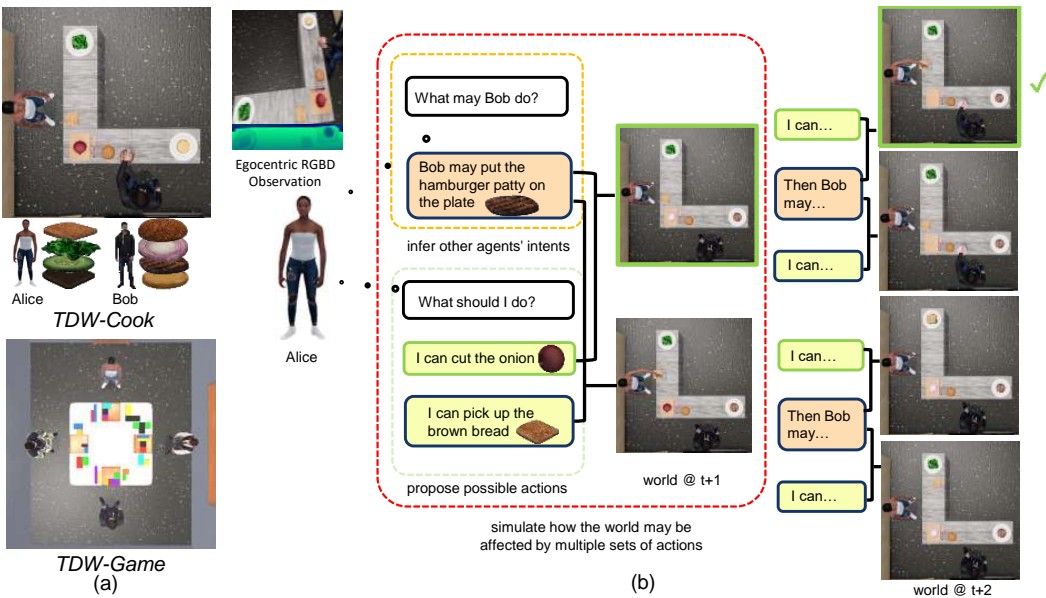

Figure 1: (a) Two challenging embodied multi-agent visual cooperation benchmarks **TDW-Cook** and **TDW-Game**, where 2-4 agents cooperate to finish dishes according to the recipe or finish puzzles according to the visual clue. (b) The agent needs to infer other agents' intents, propose possible actions, and accurately simulate how the world may be affected by multiple sets of actions to make efficient cooperation in the long run.

compositionally, which enables accurate simulation of multiple sets of actions on the world state. To address the partial observability issue, we leverage a learned generative model to estimate the overall world state given partial egocentric observations. By leveraging the compositional world model, in combination with Vision Language Models, we propose *COMBO*, a novel **C**ompositional w**O**rld **M**odel-based em**BO**died multi-agent planning framework to facilitate online cooperative planning. *COMBO* first estimates the overall world state from partial egocentric observations, then utilizes Vision Language Models to act as an Action Proposer to propose possible actions, an Intent Tracker to infer other agents' intents, and an Outcome Evaluator to evaluate the different possible outcomes. In combination with the compositional world model to simulate the effect of joint actions on the world state, *COMBO* uses a tree search procedure to integrate these modules and empowers embodied agents to imagine how different actions may affect the world with other agents in the long run and plan more cooperatively.

We instantiate the challenging embodied multi-agent visual cooperation task in ThreeDWorld (Gan et al., 2021), and build **TDW-Game** and **TDW-Cook** where 2-4 decentralized agents must cooperate to finish several puzzles according to the visual clue or dishes according to the recipe given only partial egocentric views of the world as shown in Figure 1, requiring extensive visual cooperation. The agents need to estimate the overall world state given partial egocentric observations, infer other agents' intents, and accurately simulate how the world state may be affected by multiple sets of actions to make efficient cooperation in the long run. Our extensive experiments on these two challenging benchmarks and another adapted cooperation task show our learned compositional world model delivers accurate video synthesis, conditioned on multiple sets of actions from an arbitrary number of agents and *COMBO* enables the embodied agents to cooperate efficiently with different agents across various tasks and an arbitrary number of agents. In sum, our contribution includes:

- We propose to learn a compositional world model for multi-agent cooperation by factorizing joint actions of an arbitrary number of agents and compositionally generating the video to enable accurate simulation of multiple sets of actions on the world state.

- We introduce *COMBO*, a **C**ompositional w**O**rld **M**odel-based em**BO**died multi-agent planning framework to empower the agents to imagine how different actions may affect the world with other agents in the long run and plan more cooperatively.

- We evaluate our methods on three challenging benchmarks and conduct ablation studies, the results show our framework enables the embodied agents to cooperate efficiently with different agents across various tasks and an arbitrary number of agents.

## 2 RELATED WORK

### 2.1 MULTI-AGENT PLANNING

Multi-agent planning has a long-standing history (Stone & Veloso, 2000), with various tasks and methods have been introduced (Lowe et al., 2017; Samvelyan et al., 2019; Carroll et al., 2019; Suarez et al., 2019; Jaderberg et al., 2019; Amato et al., 2019; Baker et al., 2020; Bard et al., 2020; Jain et al., 2020; Wen et al., 2022; Szot et al., 2023). For embodied intelligence, Puig et al. (2021) explored the social perception of the agents during household tasks, Zhang et al. (2023) studied the communication and cooperation of two agents in a multi-room house. However, these works didn't dig into the explicit challenge of modeling the world dynamics conditioned on an arbitrary number of agents' actions given partial egocentric observations of the world, which is essential for the agents to cooperate efficiently in long-horizon cooperation-extensive visual tasks. In contrast, we learn a compositional world model for multi-agent cooperation and propose a compositional world model-based embodied multi-agent planning framework to empower the embodied agents to imagine and plan more cooperatively with only partial egocentric observations.

### 2.2 LARGE GENERATIVE MODELS FOR EMBODIED AI

With the recent advance of large generative models (Bubeck et al., 2023; Liu et al., 2023a; Driess et al., 2023; Blattmann et al., 2023b), numerous works have explored how they can help build better agents (Wang et al., 2023b; Xi et al., 2023; Sumers et al., 2023), especially in embodied environments (Zhou et al., 2024a; Li et al., 2023; Padmakumar et al., 2022; Kolve et al., 2017; Misra et al., 2018; Xia et al., 2018; Xiang et al., 2020). Specifically, Wang et al. (2023c); Ahn et al. (2022); Sharma et al. (2021); Wang et al. (2023a); Park et al. (2023) leverage Large Language Models to help decision-making. Brohan et al. (2023); Jiang et al. (2023); Wang et al. (2023d) use vision language models to facilitate embodied agents to plan end-to-end in visual worlds. Hong et al. (2024); Black et al. (2024) use diffusion models for decision-making. Xiang et al. (2024); Du et al. (2024); Finn et al. (2016) use video models to help the robot make visual plans. There are two closely related works. VLP (Du et al., 2023b) enables a single agent to perform complex long-horizon manipulation tasks by synergizing vision-language models and text-to-video models with tree search, but they neglect the challenge of learning world dynamics conditioned on multiple sets of actions of an arbitrary number of agents. RoboDreamer (Zhou et al., 2024b) decomposes language instructions into sets of lower-level primitives and compositionally synthesizes video plans on unseen goals for robot manipulation tasks, however, they only focus on the low-level trajectory control, and assume the observation is static and non-partial. In contrast, we learn a compositional world model for multi-agent cooperation to enable accurate simulation of multiple sets of actions on the world and leverage a learned generative model to estimate the overall world state given partial egocentric observations to address the partial observability issue.

## 3 PRELIMINARIES

### 3.1 PROBLEM STATEMENT

Embodied multi-agent cooperation problem can be formalized as a decentralized partially observable Markov decision process (DEC-POMDP) (Oliehoek et al., 2016; Bernstein et al., 2002; Spaan et al., 2006), defined by $(n, S, \{A_i\}, \{O_i\}, T, G, h)$, where:

- $n$ denotes the number of agents;
- $S$ is a finite set of states;
- $A_i$ is the action set for agent $i$;
- $O_i$ is the observation set for agent $i$, including partial egocentric visual observation the agent receives through its sensors;
- $T(s, a, s') = p(s'|s, a)$ is the joint transition model which defines the probability that after taking joint action $a \in A = A_1 \times \cdots \times A_n$ in $s \in S$, the new state $s' \in S$ is achieved;
- $G$ is the final goal state;
- $h$ is the planning horizon.

Starting from an initial state $S_0 \in S$, $n$ decentralized agents need to act $a_i \in A_i$ given egocentric RGBD observations $o_i \in O_i$ each step to achieve the goal state $G$ using a minimal number of steps.

## 3.2 VIDEO DIFFUSION MODELS

Given an initial image $x_0$ and a text prompt $txt$ as the condition, a video diffusion model learns to model the distribution of possible future frames $x_{1...T}$, denote as $P_\theta(x_{1...T}|x_0, txt)$. A denoising function $\epsilon_\theta$ is trained to predict the noise applied to $x_{1..T}$ at diffusion time step $t$ given the noisy frames by optimizing the objective of

$$L_{MSE} = \|\epsilon_\theta(x_{1...T}, t|x_0, txt) - \epsilon\|^2 \tag{1}$$

where $\epsilon$ is sampled from a standard Gaussian distribution, and $t$ is a randomly sampled diffusion time step, following Ko et al. (2023).

## 3.3 COMPOSABLE DIFFUSION MODELS

Liu et al. (2022); Du et al. (2023a) shows that diffusion models are functionally similar to Energy-Based Models, and can be composed to generate images conditioned on a set of concepts $\{c_1, c_2, \cdots, c_n\}$ by training diffusion models to learn a set of score functions $\epsilon_\theta(x_t, t|c_i)$, and composing them as

$$\hat{\epsilon}(x_t, t|c_1, c_2, \cdots, c_n) = \epsilon_\theta(x_t, t) + \sum_{i=1}^{n} \epsilon_\theta(x_t, t|c_i) - \epsilon_\theta(x_t, t) \tag{2}$$

Which corresponds to the production of the probability densities

$$P_\theta(x|c_1, c_2, \cdots, c_n) \propto P_\theta(x) \prod_{i=1}^{n} \frac{P_\theta(x|c_i)}{P_\theta(x)} \tag{3}$$

Then the sampling process changes to $x_{t-1} \sim \mathcal{N}(x_t - \hat{\epsilon}(x_t, t|c_1, c_2, \cdots, c_n), \sigma_t^2 I)$.

## 4 COMPOSITIONAL WORLD MODEL

To cooperatively plan in a multi-agent scenario, in contrast to learning world dynamics in a single-agent scenario, there is an additional challenge to simulate world dynamics conditioned on the actions of an arbitrary number of agents. We propose to learn a compositional world model for multi-agent cooperation as shown in Figure 2 by factorizing the naturally composable joint actions of an arbitrary number of agents as a set of text prompts and predicting the future state by compositionally generating the future frames with a video diffusion model to enable accurate simulation of multiple sets of actions on the world state. Our compositional world model aims to model $P(s'|s, a)$, where $s$ represents the current world state, $a$ represents the joint action of $n$ agents, and $s'$ is the future world states. We first discuss how we learn the composable video diffusion model to act as a compositional world model in 4.1, then introduce the Agent-Dependent Loss Scaling we designed for the challenging situation where we need to accurately model multiple agents' manipulating multiple objects to improve the video synthesis performance in 4.2.

### 4.1 COMPOSABLE VIDEO DIFFUSION MODELS

Our compositional world model can be learned as a video diffusion model by treating the current world state $s$ as the initial frame $x_0$, and the joint actions $a$ as the text prompt, then modeling the distribution of future frames $X$ is learning the world dynamics and predicting future states $s'$. The text prompt with multiple agents interacting with multiple objects is challenging to be handled accurately. Observing that the joint action $a$ can be naturally factorized into $n$ components $a_1, \cdots, a_n$ corresponding to the action of each agent, we leverage the composable diffusion models introduced in section 3.3 to learn the compositional world model as a composition of video diffusion models conditioned on the initial frame $x_0$ and each text component $a_i$

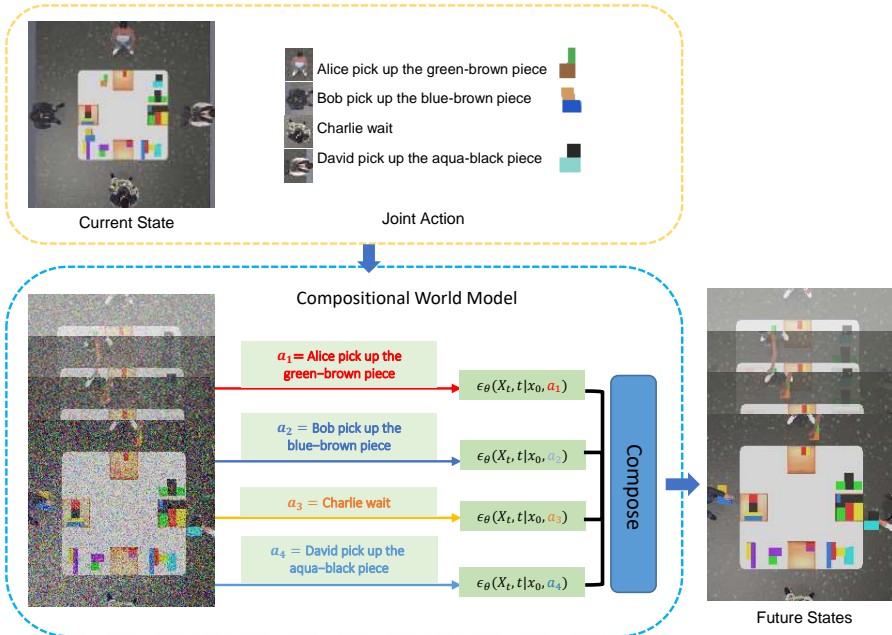

Figure 2: **Compositional World Model.** Given the current world state $x_0$ and joint action of multiple agents $a$, the compositional world model predicts the future states by first factorizing $a$ into several components $a_i$ corresponding to each agent, then generating multiple scores conditioned on the current world state and the text components, finally composing them to generate the video.

$$P(s'|s,a) = P_\theta(X|x_0,a) = P_\theta(X|x_0,a_1,\cdots,a_n) \propto P_\theta(X)\prod_{i=1}^{n}\frac{P_\theta(X|x_0,a_i)}{P_\theta(X)} \quad (4)$$

Using Equation 2, the video diffusion models learn a set of score functions $\epsilon_\theta(X_t,t|x_0,a_i)$ and composes the joint score as

$$\hat{\epsilon}(X_t,t|x_0,a) = \epsilon_\theta(X_t,t) + \sum_{i=1}^{n}\epsilon_\theta(X_t,t|x_0,a_i) - \epsilon_\theta(X_t,t) \quad (5)$$

We train this composable video diffusion model with two stages. In stage one, we learn to model the distribution of $P_\theta(X|x_0,a_i)$ by training with text component corresponding to a single agent action only using the standard denoising diffusion training objective introduced in section 3.2.

Then in stage two, we fine-tune the model to specifically learn compositional generation to model $P_\theta(X|x_0,a)$ by training with conditions containing joint actions of multiple agents using the composed score function loss

$$L_{Composed} = \|\bar{\epsilon}(X_t,t|x_0,a) - \epsilon\|^2 = \left\|\frac{1}{n}\sum_{i=1}^{n}\epsilon_\theta(X_t,t|x_0,a_i) - \epsilon\right\|^2 \quad (6)$$

After the two-stage training, we can sample from the composed distribution with the composed score function at inference time given current world state $x_0$ and joint action $a = (a_1,\cdots,a_n)$ as

$$\hat{\epsilon}(X_t,t|x_0,a) = \epsilon_\theta(X_t,t) + \sum_{i=1}^{n}\omega(\epsilon_\theta(X_t,t|x_0,a_i) - \epsilon_\theta(X_t,t)) \quad (7)$$

where $\omega$ is the guidance weight controlling the temperature scaling on the conditions.

## 4.2 AGENT-DEPENDENT LOSS SCALING

A well-trained stage one model is vital to the final compositional generation performance since the accurate modeling of $P_\theta(X|x_0,a_i)$ is the basis for modeling $P_\theta(X|x_0,a)$. We have employed a technique named *Agent-Dependent Loss Scaling* to assist in stage one training.

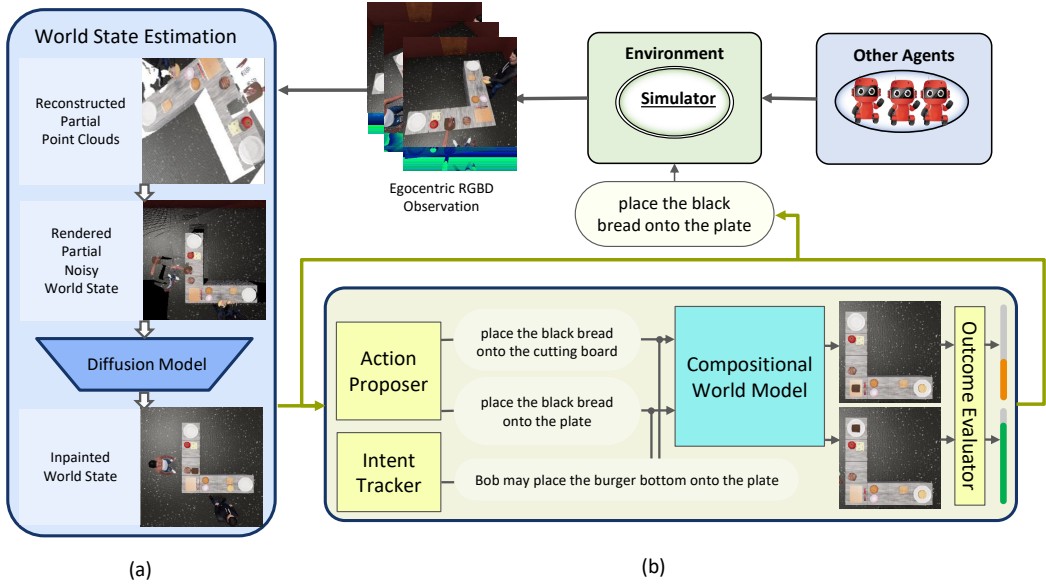

Figure 3: **Method Overview.** (a) Given partial egocentric RGBD observations, *COMBO* first reconstructs and inpaints the top-down orthographic image as the overall world state estimation. (b) *COMBO* then leverage the planning sub-modules built with Vision Language Models to propose actions, infer other agents' intents, and evaluate the outcomes simulated with the compositional world model to plan online with a tree search procedure to cooperate in the long run.

Formally, we define an agent-dependent loss coefficient matrix $C \in \mathbb{R}^{n \times H \times W}$ for $n$ agents and images of $H \times W$ pixel, and change the Equation 1 to

$$L_{MSE} = \sum_{i=1}^{n} C_i \cdot \|\epsilon_\theta(X, t|x_0, a_i) - \epsilon\|^2 \tag{8}$$

We simply set the loss coefficient matrix based on each agent's reachable region in the image to supervise the model to focus more on the related pixels. It's observed that even this straightforward loss coefficient approach brings a significant improvement to the modeling accuracy of $P_\theta(X|x_0, a_i)$.

## 5 COMPOSITIONAL WORLD MODEL FOR MULTI-AGENT PLANNING

To plan efficiently in the embodied multi-agent visual cooperation problem, we still need to address the challenge of partial egocentric observation and model complex world dynamics in the long run. We propose *COMBO*, a novel **C**ompositional w**O**rld **M**odel-based em**BO**died multi-agent planning framework, shown in Figure 3. After receiving the egocentric observations, *COMBO* first estimates the overall world state to plan on, as discussed in 5.1. *COMBO* then utilizes Vision Language Models to act as an Action Proposer to propose possible actions, an Intent Tracker to infer other agents' intents, and an Outcome Evaluator to evaluate the different possible outcomes, as detailed in 5.2. In combination with the compositional world model to simulate the effect of joint actions on the world state $s_{i+1} = CWM(s_i, a)$ introduced in 4, we discuss the tree search procedure to integrate these planning sub-modules in 5.3. *COMBO* empowers embodied agents to imagine how different plans may affect the world with other agents in the long run and plan cooperatively.

### 5.1 WORLD STATE ESTIMATION WITH PARTIAL EGOCENTRIC VIEWS

Directly planning based on partial egocentric views presents a considerable challenge. To address this, we initially reconstruct partial point clouds from multiple egocentric RGBD views $o_i$ captured from different perspectives. These point clouds are then transformed into a unified top-down orthographic image representation, serving as the world state. This representation is constructed by overlaying the views in chronological order. It is important to note that our world is inherently dynamic, with other agents actively interacting, resulting in a top-down orthographic image representation that is both noisy and incomplete, as depicted in 3 (a). To refine this representation, we employ a diffusion model to inpaint the partial and noisy orthographic image, thereby enhancing

the estimation of the world state $s_i$ for subsequent planning. This approach allows us to effectively represent and rectify the world state, enabling more accurate planning in multi-agent environments.

## 5.2 PLANNING SUB-MODULES WITH VISION LANGUAGE MODELS

**Action Proposer** Given the estimated world state $s_i$ and the long horizon goal $G$, the Action Proposer first searches over the potential action spaces $A_i$, and then proposes multiple possible actions $a_{i,1...p} = AP(s_i, G)$ to explore on. We implement this module by querying a VLM to generate the possible actions in the text given the task goal and encoded image world state as context. We finetune LLaVA on randomly collected rollouts with possible actions labeled to construct this module.

**Intent Tracker** Inferring what other agents may do given observation history is important for effective multi-agent cooperation. We implement this module $a_{-i} = IT(s_i, G)$ by keeping the estimated world state from the last $k$ steps and then feed into the VLM together with the task goal to query for the possible actions of other agents $a_{-i}$. We construct this module by finetuning LLaVA on collected short rollouts with other agents' actions labeled.

**Outcome Evaluator** It's vital to have a way to assess the value of the achieved state from different plans so the search can be effectively deepened utilizing pruning. We implement an Outcome Evaluator to fulfill this functionality $v = OE(s, G)$ by generating a heuristic score $v$ for each image state $s$ given the task goal $G$. To construct this module, we finetune LLaVA to describe the state of each object in the image and the corresponding heuristic score representing steps left to achieve the task goal considering all objects.

---

**Algorithm 1** *COMBO* Planning Procedure for Agent $i$.

---

1: **Input:** Estimated world state $s_0$ from $o_i$, task goal $G$
2: **Sub-modules:** Action Proposer $AP(s, G)$, Intent Tracker $IT(s, G)$, Compositional World Model $CWM(s, a)$, Outcome Evaluator $OE(s, G)$
3: **Parameters:** Action Proposes $P$, Planning Beams $B$, Rollout Depths $D$
4: plans $\leftarrow [[s_0]]$
5: new_plans $\leftarrow [[s_0]]$
6: **for** $d = 1 \ldots D$ **do**
7:     plans $\leftarrow$ new_plans[1...B]      # Keeps Only B Plan Beams with Best Scores
8:     new_plans $\leftarrow []$
9:     **for** plan in plans **do**
10:         $s \leftarrow$ plan[$-1$]         # Get the Last Image State in the Plan Beam
11:         $a_{i,1:P} \leftarrow AP(s, G)$     # Generate $P$ Different Action Proposals
12:         $a_{-i} \leftarrow IT(s, G)$     # Infer Other Agents' Possible Actions
13:         **for** $p = 1 \ldots P$ **do**
14:             $a \leftarrow (a_{i,p}, a_{-i})$
15:             $s_{\text{next}} \leftarrow CWM(s, a)$ # Simulate Next State Conditioned on Joint Actions
16:             new_plans.append(plan + $s_{\text{next}}$)
17:         **end for**
18:     **end for**
19:     new_plans $\leftarrow$ sorted(new_plans, $OE(s, G)$)    # Sort Plans by the Score of the Final State
20: **end for**
21: plan $\leftarrow$ new_plans[1]         # Return the Plan with the Best Score

---

## 5.3 PLANNING PROCEDURE WITH TREE SEARCH

With powerful generative sub-modules implemented so far, we can use an effective tree search algorithm to integrate them into achieving the best planning performance. Formally, we need to search for a sequence of actions that compose the plan to achieve the task goal with a minimal number of steps. To sample the long-horizon plans, we can chain the sub-modules to expand future states from current state $s_i$ with $s_{i+1} = CWM(s_i, AP(s_i, G), IT(s_i, G))$. By recursively deploying these chained modules we can expand the plans until achieving the goal state. However this is impractical due to the large search space, we implement a limited tree search procedure instead by always keeping the $B$ best-scored plan beams and exploring $P$ action proposals at each plan step for a maximum of $D$ rollout steps. We show the complete *COMBO* planning procedure in Algorithm 1.

Due to the uncertainty of other agents, *COMBO* replans at every step to reflect the changing world state. As similarly observed in (Du et al., 2023b), when searching for the best plan, the Outcome Evaluator may leverage the irregular artifacts of the Compositional World Model to get artificially

| | TDW-Game | | TDW-Cook | |
|---|---|---|---|---|
| | Cooperator 1 | Cooperator 2 | Cooperator 1 | Cooperator 2 |
| **Recurrent World Models** | 0.00 / - | 0.00 / - | 0.00 / - | 0.00 / - |
| **MAPPO** | 0.00 / - | 0.00 / - | 0.00 / - | 0.00 / - |
| **CoELA** * | 0.90 / 18.4 | 0.60 / 27.6 | 0.15 / 37.8 | 0.05 / 29.5 |
| **LLaVA** | 0.55 / 28.4 | 0.60 / 29.2 | 0.35 / 33.0 | 0.50 / 35.0 |
| **COMBO (w/o *IT*)** | 0.65 / **16.2** | 0.60 / **17.0** | 0.80 / 24.8 | 0.80 / 23.8 |
| **COMBO (Ours)** | **1.00** / 17.5 | **1.00** / 17.4 | **0.90** / **22.8** | **1.00** / 22.9 |
| Shared Belief Cooperator* | 1.00 / 15.3 | 1.00 / 15.9 | 0.95 / 24.0 | 1.00 / 21.4 |

Table 1: **Main results on TDW-Game and TDW-Cook.** We report the mean success rate over a horizon of 30 and the average steps of the successful episodes over 20 episodes here. *COMBO* (w/o *IT*) denotes *COMBO* without the Intent Tracker module. **Shared Belief Cooperator** has access to the Oracle state of the world and other agents' policies. * denotes Oracle vision perception.

| 2D-FetchQ.* | BC-single | BC-GAIL | Co-GAIL | COMBO (Ours) |
|---|---|---|---|---|
| Success Rate | 21.1 | 27.2 | 53.3 | **81.3** |

Table 2: **Results on 2D-FetchQ.** We report the success rate on replay evaluation over 60 episodes.

high scores, such as where key objects are teleported to desired positions. To mitigate this artificial exploitation issue, the Outcome Evaluator will use the default score from the last state if the new state generated from the Compositional World Model suspiciously increases the score above a threshold.

# 6 EXPERIMENTS

## 6.1 EXPERIMENTAL SETUP

We instantiate the challenging embodied multi-agent visual cooperation task in ThreeDWorld and build two challenging benchmarks: **TDW-Game** and **TDW-Cook** where 2-4 decentralized agents must cooperate to finish several puzzles according to the visual clue or dishes according to the recipe given only egocentric views of the world as shown in Figure 1, requiring extensive visual cooperation. All the agents have an observation space of egocentric $336 \times 336$ RGBD images and the corresponding camera matrix. We also adapt the **2D-FetchQ** challenge from Wang et al. (2022) with a visual observation space and high-level action space to evaluate our method. In this challenge, two agents must coordinate their strategy to hold buttons to unlock rooms and fetch treasures from the unlocked room. More details on the tasks can be found in Appendix A. We also verify our method in a real-world human-robot cooperation task in Appendix D.

**Metrics** We evaluate the agent's cooperation ability by task success rate and average steps of the successful episodes when cooperating with different agents across 20 episodes. An episode is done when the goal state is reached and considered successful, or the maximum planning horizon $h$ is reached and deemed failed. Specifically, we implement two different agent policies as cooperators on both tasks to reliably evaluate the agents, which are elaborated in Appendix A.1.1.

**Baselines** We compare our methods to several multi-agent planning methods:
- Recurrent World Models (Ha & Schmidhuber, 2018), using VAE to encode the visual observations and a generative recurrent neural network to learn the world model. An evolution-trained simple policy is used as a controller.
- MAPPO (Yu et al., 2022), a MARL baseline where agents with shared weight Actor and Critic are trained jointly with PPO.
- CoELA (Zhang et al., 2023), an LLM agent with access to the Oracle vision perception, re-implemented due to the lack of ground truth segmentation needed to train the Perception Module.
- LLaVA (Liu et al., 2023a), an end-to-end VLM (LLaVA v1.5 7B) agent fine-tuned with the same collected rollouts to train the Action Proposer in our method.
- Shared Belief Cooperator, a planner that shares the same policy as the cooperators and has access to the Oracle state of the environment and other agents' policies, acting as a strong baseline.

**Implementation details** The video diffusion model of the compositional world model is built upon AVDC (Ko et al., 2023) codebase, with the architectural modification of introducing a cross attention layer to the text condition in the ResNet block and replacing the Perceiver with an MLP to enhance the text conditioning. More details are in Appendix B.

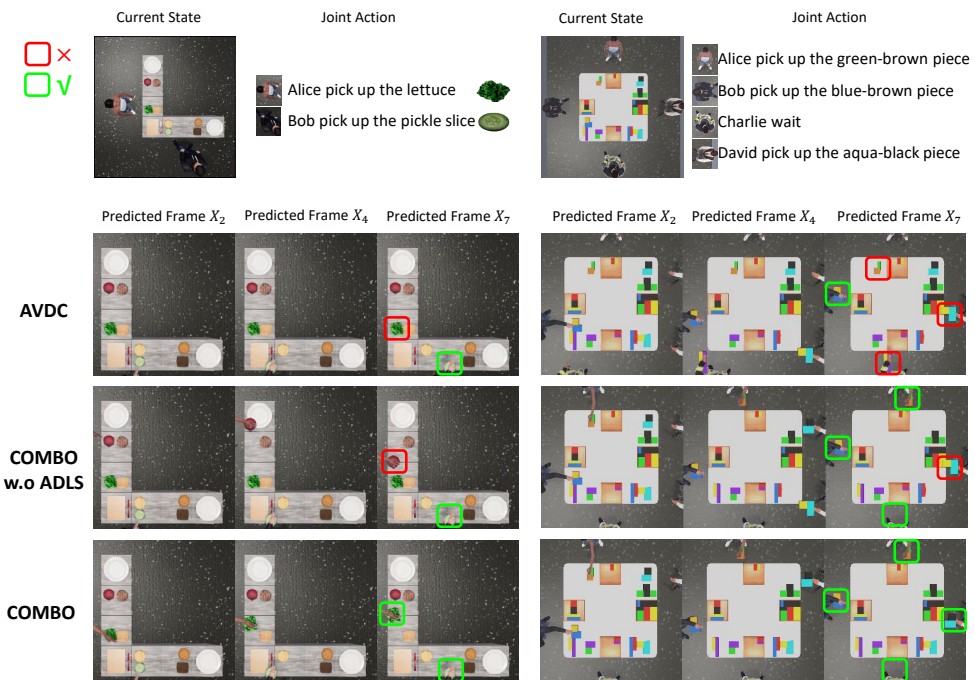

Figure 4: **Compositional World Model learns world dynamics better.** Our compositional world model can simulate world dynamics conditioned on the joint action of multiple agents accurately while **AVDC** struggles with simulating which agents should act, and **COMBO w.o ADLS** may simulate actions incorrectly.

| | TDW-Game | | | TDW-Cook | | |
|---|---|---|---|---|---|---|
| | **Single** | **Multiple** | **Plan** | **Single** | **Multiple** | **Plan** |
| **AVDC** | 65% | 20% | 29.7(80%) | 85% | 25% | 34.5(90%) |
| **COMBO w.o *ADLS*** | 70% | 55% | 26.9(100%) | 85% | 70% | 28.3(100%) |
| **COMBO** | **95%** | **75%** | **17.5(100%)** | **100%** | **100%** | **21.5(100%)** |

Table 3: **Human evaluation of generated videos from different World Models and the corresponding plan performance**. **Single** denotes the accuracy of synthesized videos conditioned on a single agent's action among 20 samples. **Multiple** denotes the accuracy of synthesized videos conditioned on multiple agents' actions among 20 samples. **Plan** denotes the average steps (success rate) across cooperating with two different cooperators over 10 episodes. Best performances are in **bold**.

## 6.2 RESULTS

***COMBO* can cooperate efficiently with different cooperators** As shown in Table 1, both cooperators of different policies achieve the best performance when cooperating with *COMBO* on **TDW-Game** and **TDW-Cook**, finishing all the tasks within the limited horizon with the least steps. Both Recurrent World Models and MAPPO perform poorly due to the difficulty of the multi-agent long-horizon task with only egocentric observations. Compared to LLM Agent CoELA and VLM Agent LLaVA, *COMBO* with a compositional world model to explicitly model the world dynamics and plan accordingly achieves more efficient cooperation. On adapted **2D-FetchQ**, COMBO surpasses other behavior cloning or Co-Gail baselines in a more challenging setting as shown in Table 2, demonstrating the efficacy and adaptability of our proposed method.

**Intent Tracker contributes to efficient cooperation** Comparing the results of *COMBO (w.o IT)* and *COMBO* in Table 1, we can see the Intent Tracker is important to empower the agent to infer other agents' intents and consider them during planning for better cooperation in the long run. A qualitative case demonstrating the adaptability of the Intent Tracker Module is shown in Appendix C.1.

**Compositional World Model is crucial to make cooperative plans** We compared the generated video quality with AVDC (Ko et al., 2023), a video diffusion model designed for robotics tasks, and ablate on the effect of the Agent-Dependent Scaling Loss (COMBO w.o. *ADSL*) in Table 3 and Figure 4. Compared to AVDC, our CWM can accurately simulate which agents are acting and what interactions are described in the text condition, leading to a large performance gain in the

| Action Proposals $P$ | Rollout Depth $D$ | Plan Beam $B$ | Success Rate | Average Steps |
|---|---|---|---|---|
| 3 | 1 | 1 | 50% | 27.8 |
| 2 | 2 | 2 | 70% | 18.1 |
| 3 | 3 | 3 | 100% | 17.5 |

Table 4: **Plan performance improves with more compute budgets.** We report the mean success rate and average steps of the successful episodes cooperating with two different cooperators over 5 episodes on **TDW-Game** here.

| | *4-agent* | *3-agent* | *2-agent* |
|---|---|---|---|
| **LLaVA** | 0.58 / 28.7 | 0.65 / 27.4 | 0.80 / 20.7 |
| **COMBO** | **1.00 / 17.5** | **1.00 / 15.0** | **1.00 / 10.5** |
| Shared Belief* | 1.00 / 15.9 | 1.00 / 13.5 | 1.00 / 8.7 |

Table 5: **Results on TDW-Game with different number of agents.** We report the mean success rate over a horizon of 30 and average steps cooperating with two different cooperators over 10 episodes here.

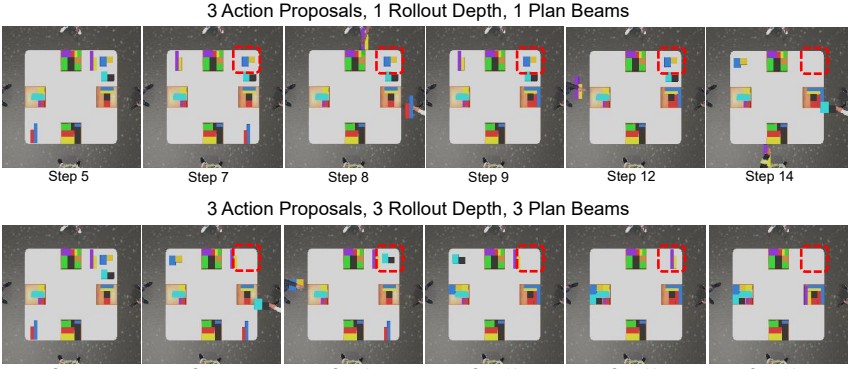

Figure 5: **More Computation budgets leads to better plan**. With more computation budgets (second row), *COMBO* can search for a better plan where Alice first clears the common region with David so that he can pass the next puzzle piece to her instead of having to wait, leading to a better state after same number of steps.

accuracy of modeling agent actions on the world state. Removing the Agent-Dependent Scaling Loss leads to considerable accuracy degradation, showing the effectiveness of the introduced technique. Moreover, with the compositional world model, the overall plan performance boosts to 17.5 steps for **TDW-Game** compared to 29.7 steps with AVDC. See C.3 for additional failure case analysis.

**More computation budgets lead to better Plan** We study the effect of computation budgets on the long-horizon plan quality in Table 4, where we can see more computation budgets with more action proposals, larger plan beams, and deeper rollout depth leads to higher success rate and less average steps. A qualitative example of generated plans with different computation budgets is shown in Figure 5, where more searches can obtain better plans in the long run.

***COMBO* can generalize to an arbitrary number of agents** *COMBO* trained with only data of four agents playing **TDW-Game** can surprisingly generalize to three and two agents version well as shown in Table 5, showing the strong generalization and promising future of the compositional world model-based modularized planning framework for multi-agent cooperation.

# 7 CONCLUSION

In this work, we learn a compositional world model for embodied multi-agent cooperation by factorizing the naturally composable joint actions of an arbitrary number of agents and compositionally generating the video to enable accurate simulation of multi-agent world dynamics. We then propose ***COMBO***, a novel Compositional World Model-based embodied multi-agent planning framework to empower the agents to infer other agents' intents and imagine different plan outcomes in the long run. Our experiments on three challenging embodied multi-agent cooperation tasks show ***COMBO*** can enable the embodied agents to cooperate efficiently with different agents across various tasks and an arbitrary number of agents.

**Future Work** Our method leverages tree search combined with Large Generative Models to devise long-horizon plans. The necessity for multiple inferences using large models leads to a relatively slow inference speed, restricting the applicability of our method in scenarios demanding rapid response. Exploring the development of more efficient models could potentially mitigate this drawback and enhance the practicality of our approach in time-sensitive environments.

ACKNOWLEDGMENTS

We thank Jeremy Schwartz and Esther Alter for setting up the ThreeDWorld environments, and Chunru Lin for the help with real-world experiments. This project is supported by the Honda Research Institute.

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

# A ADDITIONAL EXPERIMENT DETAILS

## A.1 TDW TASKS

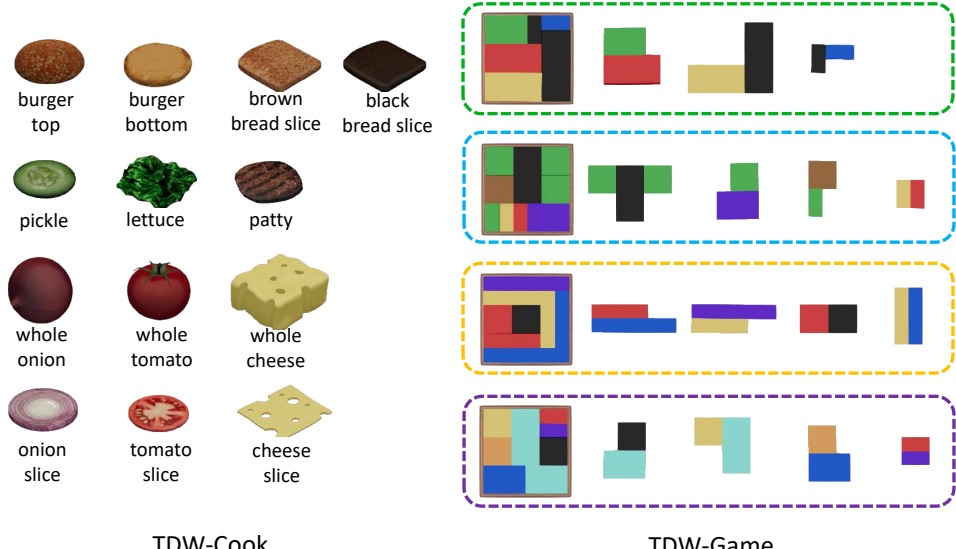

Figure 6: **Objects of TDW-Cook and TDW-Game.** Left: 13 food items that may occur on the table while 10 of them (except for the third row) may occur in the recipe. Right: 4 puzzles containing 3 to 4 pieces each with different colors and shapes.

We instantiate the challenging embodied multi-agent visual cooperation task in ThreeDWorld (Gan et al., 2021), and build two challenging benchmarks: **TDW-Game** and **TDW-Cook** where 2-4 decentralized agents must cooperate to finish several puzzles according to the visual clue or dishes according to the recipe given only partial egocentric views of the world.

In **TDW-Game**, 3-4 agents cooperate to pass and place 6-8 puzzle pieces scattered randomly on the table into the correct puzzle box according to visual clues such as the shape. The location of the puzzle box and pieces are randomly initialized across different episodes. The action space includes *wait, pick up [obj], place [obj] onto [loc]*.

In **TDW-Cook**, 2 agents cooperate to pass, cut, and place 6-8 out of 10 possible food items randomly scattered on the table to make some dishes according to the recipe. The location of the food items and the recipe are randomly initialized across different episodes. Specifically, there is only one cutting board where agents can pass or cut objects, making cooperation vital for efficient plans. The action space is the same as **TDW-Game** with an addition of *cut [obj]*.

The object assets of the two tasks are illustrated in Figure 6.

### A.1.1 COOPERATOR POLICIES

In **TDW-Game**, Cooperator 1 takes the policy of always passing the unwanted puzzle pieces in a clockwise manner, while Cooperator 2 takes the policy of always passing the unwanted puzzle pieces in a counter-clockwise manner.

In **TDW-Cook**, Cooperator 1 takes a "selfish" policy of always operating objects in its own recipe first, while Cooperator 2 takes an "altruistic" policy of always prioritizing operating objects in the cooperator's recipe.

### A.1.2 BASELINES

**MAPPO** Our task is multi-agent reinforcement learning with decentralized observation and control, as different agents have disjoint observations to produce their actions. We followed the training procedure in Yu et al. (2022). Each agent is an actor-critic network where the actor and critic share

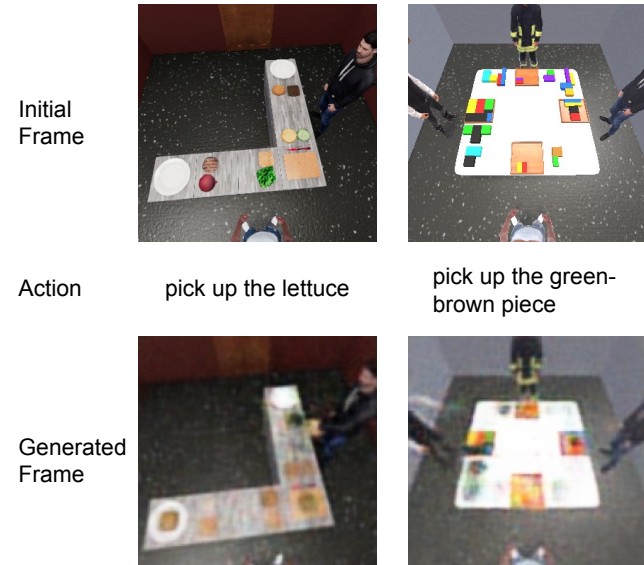

Figure 7: **Recurrent world models failed to model world dynamics conditioned on single actions accurately.**

a common convolutional backbone. We pre-define a set of all possible actions for each environment and let the PPO agent choose among them. We design rewards to reflect the steps left to finish the episode in a similar spirit to the Outcome Evaluator's heuristics score design.

However, in our setting, the embodied multi-agent simulation samples are relatively slow, creating a higher demand for sample efficiency of the method. We conducted a grid search on hyper-parameters of learning rate in {1e-6, 5e-5, 1e-3}, batch size in {2, 64}. However, the results are poor and the behavior either falls to near-random or collapses to a single action, which verifies the challenge of cooperative planning on ego-centric RGBD observations.

**Recurrent World Models**   We followed the training procedure in Ha & Schmidhuber (2018), firstly training a variational autoencoder (VAE) on the same data collected for training *COMBO* to encode the egocentric observations in the **TDW-Game** and **TDW-Cook** environments into a 32-dimensional latent vector $z \in \mathbb{R}^{32}$. Then, we train a mixture density network combined with a recurrent neural network (MDN-RNN) for predicting the future latent vector given a textual action which is encoded by BERT (Devlin et al., 2019), the current egocentric observation and the current hidden state of the RNN, which is modeling $P(z_{t+1}|a_t, z_t, h_t)$. Finally, we train two simple controllers using the covariance-matrix adaptation evolution strategy (CMA-ES) (Hansen, 2016) to maximize the expected cumulative reward of a rollout by interacting with the actual environments.

The recurrent world model failed to model world dynamics accurately conditioned on a single action, as shown in Figure 7. Specifically, we predict the latent code $z_{t+1}$ from MDN-RNN and use the decoder of the VAE to generate the image of the next time step. Additionally, the simple controller doesn't work well on a task that requires complex planning as it keeps collapsing to single actions in our experiments.

### A.2   2D-FETCHQ TASK

As shown in Figure 8, the **2D-FetchQ** environment from Wang et al. (2022) was originally inspired by the Fetch-Quest collaborative game in SuperMario Party where two agents must coordinate their strategy to hold buttons to unlock rooms and fetch treasures from the unlocked room. At the beginning of the game, two treasures are locked in the rooms located in two corners of the map. For each agent, the only way to fetch a treasure is to let its collaborator press the button beside the room to hold the door. After the collaborator presses the button, the agent can enter the room and get the treasure. To win the game, both agents should get a treasure and reach the closest destination.

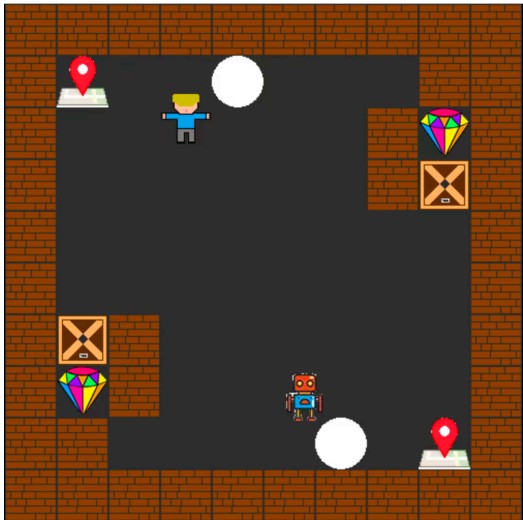

Figure 8: **2D-FetchQ Task.** Two agents coordinate their strategy to hold buttons to unlock rooms and fetch treasures from the unlocked room.

Therefore, the agent who has already gotten the treasure will help the other agent to fetch its treasure. We adapted this environment with a visual observation space and high-level action space to evaluate our method.

**Observation Space**    $128 \times 128$ RGB image.

**Action space**    *move up*, *move down*, *move lfet*, *move right*, *wait*.

**Baselines**    **BC-single** refers to directly applying behavior cloning to learn robot trajectories. **BC-GAIL** involves using behavior cloning to learn human trajectories first, followed by interactive learning of robot trajectories through GAIL (Generative Adversarial Imitation Learning) with the human policy. **Co-GAIL** leverages data of human-human collaboration demonstrations as guidance to concurrently generate simulated interactive behaviors and train a human-robot collaborative policy.

**Evaluation**    We evaluated COMBO using **replay evaluation**, where COMBO collaborates with human demonstration trajectories collected through replay and determines whether the task succeeds.

## B    ADDITIONAL IMPLEMENTATION DETAILS

### B.1    COMPOSITIONAL WORLD MODELS

**Implementation Details**    We use the base resolution $128 \times 128$ for the video diffusion model and train a super-resolution diffusion model to get the final $336 \times 336$ images. We use the T5-XXL encoder to pre-process all the text conditions for a better contextual representation. All vision language models used in the main experiments are LLaVA-v1.5-7B. The planning parameters are set to Action Proposes P=3, Planning Beams B=3, and Rollout Depths D=3 unless other specified.

**Dataset Collection**    We collect random rollouts with a scripted planner and generated 107k videos of **TDW-Game** and 50k videos of **TDW-Cook**.

**Model Parameters**    The parameters of our inpainting diffusion model, super-resolution diffusion model, and video diffusion model are shown in Table 6.

|  | Inpainting | Super-Resolution | Video Diffusion Model |
|---|---|---|---|
| **num_parameters** | 84M | 38M | 198M |
| **input_resolution** | $336 \times 336$ | $128 \times 128$ | $128 \times 128$ |
| **output_resolution** | $336 \times 336$ | $336 \times 336$ | $128 \times 128$ |
| **base_channels** | 64 | 64 | 128 |
| **num_res_block** | 2 | 2 | 2 |
| **attention_resolutions** | (16,) | (16,) | (8, 16) |
| **channel_mult** | (1, 2, 4, 6, 8) | (1, 2, 3, 4, 5) | (1, 2, 3, 4, 5) |

Table 6: **Model Parameters.**

**Agent-dependent Loss Scale**  We use a simple method to set the loss scale. If agent $i$ is located in the upper half of the image, then we set $C_{i,1:H/2,1:W} = 2$ and others are 1. Similarly, the same approach is applied to other situations or conditions.

We utilize the agents' reachable regions to define the coefficient matrix here, but it's not the only approach. The key idea of this technique is to encourage the model to focus more on the pixels related to the action prompt. There are also other possible ways to leverage this technique when such information of reachability is not available. One promising way is to use advanced object detection and segmentation models like GroundingDINO (Liu et al., 2023b) and SAM2 (Ravi et al., 2024) to identify the relevant regions in an image (e.g., "the lettuce" in the action prompt "Bob pick up the lettuce") and adjust the loss scale for those regions. Another promising approach is to discover reachability during exploration, such as bootstrapping from zero-shot vision-language models (VLMs), where an agent could learn reachability by interacting with the environment.

**Compute**  We train the world model for 50k steps in the first stage with a batch size of 384 on 192 V-100 GPUs in 1 day. Then, we fine-tune the model for 25k steps in the second stage with 120 batch size on 120 V-100 GPUs in 1 day. Both the inpainting model and the super-resolution model are trained for 60k steps with a batch size of 288 on 24 V-100 GPUs in 1 day.

**Samping**  We use DDIM sampling across the experiments with guidance weight 5 for the text-guided video diffusion model.

### B.2  PLANNING SUB-MODULES

**Training Data Collection**  We generated 4k training environments by randomly sampling a recipe in TDW-Cook and a puzzle box in TDW-Game for each agent. All objects were initialized at random positions on the table. Scripted planners with randomness were then used to play out the episodes and collect the state and action histories. We finetune LLaVA-1.5-7B (Liu et al., 2023a) with LoRA (Hu et al., 2021) for one epoch to obtain one shared model for each submodule across all tasks and cooperators.

For the Intent Tracker, we collected 40k short rollouts consisting of three images of consecutive observations and a textual description of the next actions of all the agents, converted by a template given the action history.

For the Outcome Evaluator, we collected 138k data consisting of one image of the observation and a textual description of the state of each object in the image and the heuristic score, converted by a template given each object's location.

**Compute**  For each planning sub-module, we finetune LLaVA-1.5-7B with LoRA for one epoch with a batch size of 144 on 18 V-100 GPUs in about 3 hours.

# C    ADDITIONAL RESULTS AND DISCUSSIONS

## C.1    INTENT TRACKER CAN ADAPT THE PREDICTIONS

The intent tracker module should generalize to unseen cooperators and adapt its predictions dynamically based on historical observations. Our approach employs a shared intent tracker module across all tasks and cooperators. The test-time cooperator's policy is agnostic to our agent during training, meaning the intent tracker must rely on observation history to adapt its predictions and track the intents of diverse cooperators effectively. As shown in Figure 9, given the same current state, the intent tracker accurately predicts the next action of Agent Bob by leveraging its observation history, which reveals a preference for preparing his recipe first or helping Agent Alice first. With accurate intent tracking, it becomes possible for Agent Alice to select the best action to take.

While the intent tracker module is trained on a limited set of scripted planners (e.g., the 4-agent TDW-Game setting), it generalizes reasonably well to new behaviors due to its reliance on cooperator-independent observation patterns. As demonstrated in Table 5, the same intent tracker was applied without retraining in 3-agent and 2-agent versions of TDW-Game, where cooperators followed policies unseen during the training phase. The tracker demonstrated adaptability, achieving correct intent prediction rates of $75.8\%$ in the 2-agent cooperation scenario and $64.6\%$ in the 3-agent cooperation scenario. Among the wrong predictions, $20\%$ attributes to the intent prediction of "wait" while it shouldn't be. This adaptation showcases the model's ability to adapt dynamically and predict intents effectively without over-reliance on memorization.

Generalizing to more diverse and complex cooperator behaviors remains a challenging yet promising direction for future work. Enhancements such as Population-Based Training (Jaderberg et al., 2019) and Behavior Diversity Play (Szot et al., 2023) could expand the variety of cooperators encountered during training, thereby increasing the robustness of the intent tracker. Furthermore, integrating advanced modeling approaches, such as the Bayesian Theory of Mind (Wu et al., 2021), could facilitate more nuanced inference and adaptability in multi-agent collaboration scenarios.

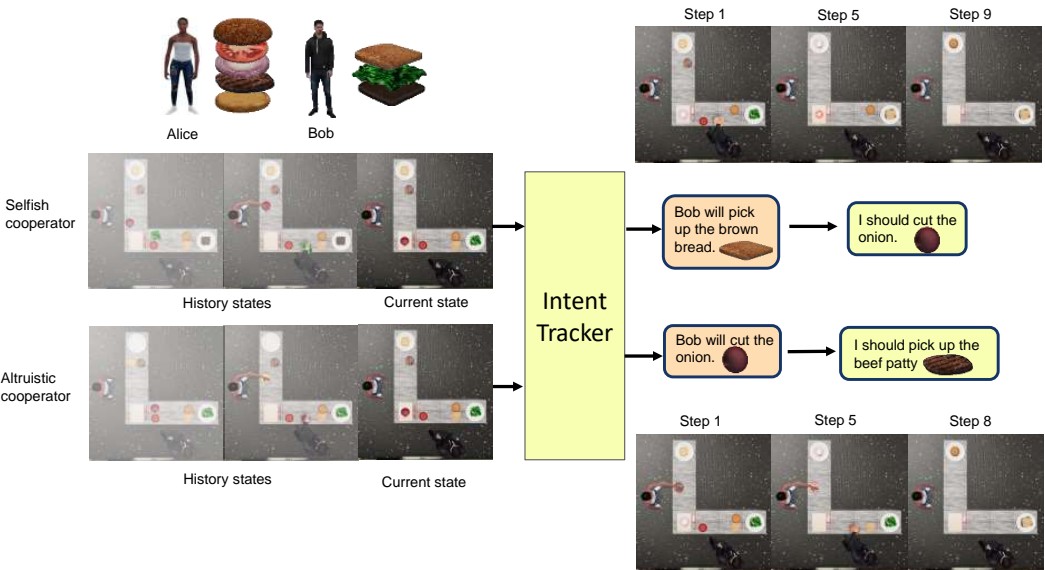

Figure 9: **The Intent Tracker Module dynamically adapts its predictions based on historical observations.** When collaborating with agents exhibiting either selfish or altruistic policies, the Intent Tracker accurately predicts the cooperator's next action by considering their historical behavior. This enables the agent to determine its appropriate next action based on the predicted intent, even when faced with the same current state but different historical contexts.

|  | LLaVA-1.5-7B | Simpler Supervised Model |
|---|---|---|
| **Action Proposer** | **98.3** | 90.3 |
| **Intent Tracker** | **79.4** | 73.2 |
| **Outcome Evaluator** | **99.7** | 62.7 |

Table 7: **Comparison of LLaVA-1.5-7B and the Simpler Supervised Model across different components.** The accuracy of the model prediction on the test set.

## C.2 OTHER IMPLEMENTATION ALTERNATIVES

These planning sub-modules could also be implemented with other methods. Additionally, we implemented a simpler supervised model to serve as the planning sub-modules and trained it using the same dataset we employed for fine-tuning the VLM. This model encodes the input image using a *VIT-B/16 CLIP* encoder and the textual task description using the *BERT-base-uncased* encoder. The encoded outputs are concatenated and passed through a two-layer fully connected network with a hidden dimension of **256**.

- For the Action Proposer, the output is a multi-hot vector of 157 dimensions, representing all possible actions.

- For the Intent Tracker, the output is a vector of $4 \times 157$ dimensions, capturing intents across agents.

- For the Outcome Evaluator, the output is a scalar representing the heuristic score.

**Compute** For each component, we preprocess the images and the textual prompts to extract the features on 8 V100 GPUs in about 30 minutes. We then train the supervised model for 10 epochs with a batch size of 64 on 1 V100 GPU in about 20 minutes.

For each component, we selected the best result across 10 training epochs. The results shown in Table 7 indicate that VLMs not only offer better generalizability but also achieve superior performance compared to this supervised model. This can be attributed to the common-sense knowledge embedded in the VLM during its pretraining on large-scale internet data, as well as its superior ability to integrate the two modalities effectively.

## C.3 FAILURE CASE ANALYSIS

As shown in Table 3, the compositional world model still fails 25% of the time for predicting the next state conditioned on four agents' actions. We attribute the errors to two primary factors:

- **Image Quality:** This is the most critical issue, accounting for 80% of the errors. We believe this problem arises from the use of CFG (Classifier-Free Guidance) based on multiple textual action conditions. Since CFG strongly guides the diffusion model to fulfill textual action conditions, it often results in contradictory behaviors, such as an agent simultaneously placing a block on both the left and right sides (see Figure 10) or attempting to pick up a block but failing to do so.

- **Misunderstanding:** This error type involves the agent failing to act according to the textual action conditions (see Figure 11). Compared to image quality issues, misunderstanding errors are less frequent, likely due to the influence of CFG. These errors may stem from data imbalance, insufficient training data, or inadequate model generalization capabilities.

Incorporating hidden physical parameters, such as object mass, is an essential direction for making the task closer to real-world scenarios. Extending our framework to include physics-aware video generation approaches (Liu et al., 2024; Zhang et al., 2024), is a promising avenue for future work and could further improve the capability of world models to handle hidden variables effectively.

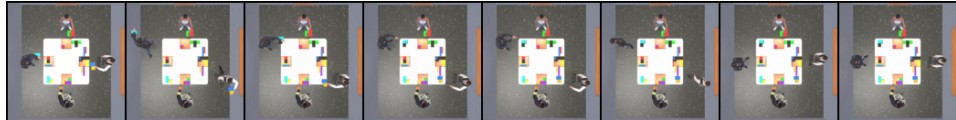

Figure 10: **An example of the image quality error.** The agent at the bottom places the green-black piece on both sides.

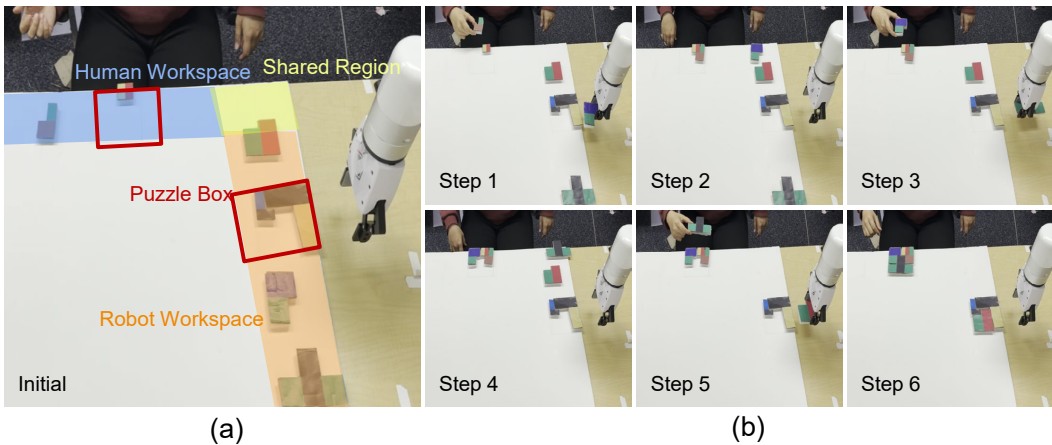

Figure 11: **An example of the misunderstanding error.** The textual action condition for the agent on the right is to "place the blue-brown piece onto the right border of the reachable region", but the agent places the piece on the left side.

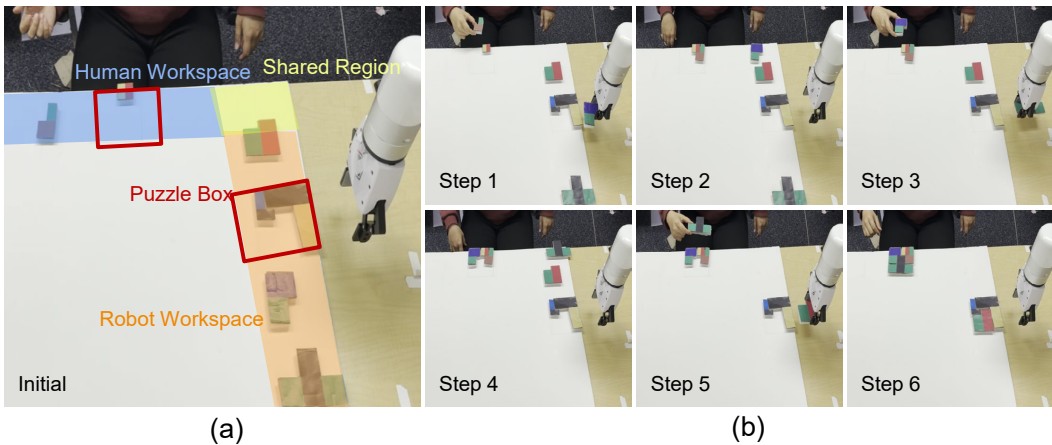

Figure 12: (a) **Real-world task set up.** The reachable region of the human and the robot is shown in blue and yellow respectively. Their shared region is shown in green, and the puzzle box is shown in red. They need to cooperate closely to pass and finish the puzzles. (b) **An example plan execution.** The robot passes the pieces the human needs first and then finishes its own puzzle.

## D  REAL-WORLD TASK: HUMAN-ROBOT COLLABORATION

**Task Setup**: We instantiate the TDW-Game environment in the real world with two agents. As shown in Figure 12(a), a human and an XArm sit on the upper and right sides of the table, each can only reach the objects on his side of the table, so there is only one shared region to exchange objects, which is located at the upper-right corner of the table. They must cooperate to pass and place 3-5 puzzle pieces scattered randomly on the table into the correct puzzle box according to visual clues such as the shape. In each step, the human or the robot can only pick or place one object within his reachable region.

**Results**: We fine-tune the Compositional World Model with collected real-world data and use the same VLMs without fine-tuning for other sub-modules. We conducted 5 trials of experiments and reported the number of successful trials and the average steps in Table 8. An example plan execution is shown in Figure 12(b), where the robot first helps pass the puzzle pieces the human needs, and finally picks and places its last piece of the puzzle to finish the task. We also show two sampled video generations of the compositional world model from the same current state and conditioned on two different joint actions in figure 13. The result demonstrates our method is capable in real-world scenarios involving low-level control and non-trivial generalization.

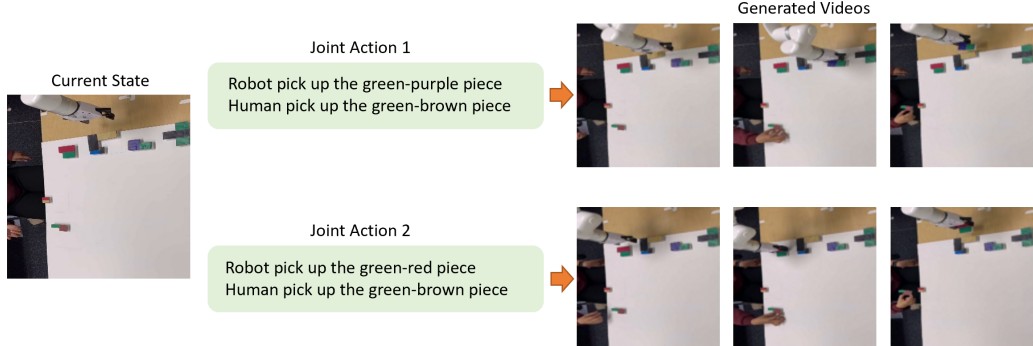

Figure 13: **Two sampled video generations of the compositional world model conditioned on the same current state but different joint actions.**

|  | **Success** | **Average Steps** |
|---|---|---|
| *COMBO* | 4/5 | 8.4 |

Table 8: **Real-world experiment results.** We report the number of successful trials / total trials and the average steps of the successful trials over 5 trials here.

