# OpenReview forum: "COMBO: Compositional World Models for Embodied Multi-Agent Cooperation"
_ICLR.cc/2025/Conference — ICLR 2025 Poster_

### Official Review · Reviewer_Dcsa · 2024-10-24

**Soundness:** 3
**Presentation:** 3
**Contribution:** 3
**Rating:** 6
**Confidence:** 3

**Summary:**

The authors of this paper explored a compositional workload model for embodied multi-agent cooperation. Unlike previous approaches to embodied multi-agent cooperation, they introduced an explicit world model that simulates the next state by incorporating the estimated actions from the Intent Tracker along with the agent's actions. This model stands out from other single-agent, model-based reinforcement learning approaches by explicitly accounting for multi-agent cooperation dynamics. Utilizing this world model, they implemented a tree search planning strategy similar to Monte-Carlo Tree Search [1]. Their approach demonstrated superior performance in embodied multi-agent cooperation tasks and, due to the compositional nature of their model, exhibited strong generalization capabilities.

[1] Silver, David, et al. "Mastering the game of Go with deep neural networks and tree search." nature 529.7587 (2016): 484-489.

**Strengths:**

- They designed explicit compositional world model. It is one of the most distinguishable designs in their modeling, which supports the look ahead planning (tree search planning), and they showed it is beneficial through the empirical evaluation results. As a part of this, the world state estimation is good to build the world model for multi-agent setting.
- It is a well written paper. I can easily follow their discussions without unnecessary questions.

**Weaknesses:**

- The room of the evaluated benchmarks is too small to show the effectiveness of their proposed modeling. In Table 1, COMBO outperformed previous works. Although, when comparing with LLaVA, it shows comparable success rate except TDW-Cook Cooperator 1 setting. Their efficiency on solving the tasks is clearly better than LLaVA, but we felt it is not good enough to evaluate the effectiveness of their modeling.
- The generalization performance evaluation is too weak to show that in lines 521-524 and Table 5. They trained the model with 4 agents and applying it to 3 agent only, then what happens if we test with 2 or much more agents such as 10? Additionally, COMBO is more efficient than LLaVA, but LLaVA also showed good success rates for 3 agents setting. The LLaVA is trained on the 3 agents setting? If yes, it should be mentioned in the caption of the figure or the paragraph. If no, then we think this empirical evidence is weak to show the generalizationality of the COMBO.

**Questions:**

- In Figure 1, what means the surrounding figures? Maybe we guess it is to visualize the look ahead planning, but I am not sure it is a good visualization for showing that.

---

> ### Author Response · Authors · 2024-11-24
> **Response to Reviewer Dcsa**
>
> *We sincerely thank the reviewer for the time to read our paper, and we appreciate your positive and constructive comments! We address your questions in detail below and updated our paper according to your suggestions.*
>
> > Q1: In Table 1, COMBO outperformed previous works. Although, when comparing with LLaVA, it shows comparable success rate except TDW-Cook Cooperator 1 setting. Their efficiency on solving the tasks is clearly better than LLaVA, but we felt it is not good enough to evaluate the effectiveness of their modeling.
>
> Thanks for pointing this out! We agree that the current success rate metric does not provide enough detailed information for comparing the baselines. To address this, we have introduced an additional metric **success rate@30**, which evaluates the success rate with a shorter planning horizon of $h=30$. The results, shown in the table below, highlight that our method **COMBO clearly outperforms other baselines, including LLaVA**, achieving an average success rate of **0.98** compared to **0.5** for the LLaVA baseline.
>
>
> || TDW-Game Cooperator 1 | TDW-Game Cooperator 2 | TDW-Cook Cooperator 1 | TDW-Cook Cooperator 2 |
> |---------------------------|------------------------|------------------------|------------------------|------------------------|
> | **CoELA \***              | 0.90                  | 0.60                  | 0.15                  | 0.05                  |
> | **LLaVA**                 | 0.55                  | 0.60                  | 0.35                  | 0.50                  |
> | **COMBO (w/o IT)**        | 0.65                  | 0.60                  | 0.80                  | 0.80                  |
> | **COMBO (Ours)**          | **1.00**                  | **1.00**                  | **0.90**                  | **1.00**                  |
> | **Shared Belief Cooperator \***  | 1.00                  | 1.00                  | 0.95                  | 1.00                  |
>
> - The higher success rate of LLaVA with a longer planning horizon can be attributed to the randomness in its predictions, which occasionally allows it to escape from dead loops when given sufficient steps.
>
> > Q2: The generalization performance evaluation is too weak to show that in lines 521-524 and Table 5. They trained the model with 4 agents and applying it to 3 agent only, then what happens if we test with 2 or much more agents such as 10? Additionally, COMBO is more efficient than LLaVA, but LLaVA also showed good success rates for 3 agents setting. The LLaVA is trained on the 3 agents setting?
>
> Thanks for your valuable question. Following your suggestions, We **additionally tested our method and the LLaVA baseline on the 2-agent version of TDW-Game without further training**. The success rate@30 and average steps when cooperating with two different cooperators over 10 episodes are reported in the table below. The results demonstrate that our method **COMBO clearly outperforms LLaVA across all settings of different numbers of agents**, successfully completing all tasks within 30 steps without requiring additional training. These results have also been incorporated into Table 5 of the manuscript.
>
> |Success Rate@30, Steps | 2-agent | 3-agent | 4-agent |
> | -- | --- | --- |---|
> LLaVA | 0.8, 20.7 | 0.65, 27.4 | 0.58, 28.7
> COMBO | **1.0, 10.5** | **1.0, 15.0** | **1.0, 17.5**
>
> - Although the LLaVA baseline was also not further trained, it benefits from being pretrained on internet-scale data and only needs to propose an action for itself, which makes it inherently easier to transfer to new scenarios. On the other hand, COMBO includes a world model trained entirely from scratch using only the 4-agent TDW-Game trajectories. It is therefore remarkable that COMBO demonstrates strong generalization to the 3-player TDW-Game, despite the significantly different agent dynamics in this setting, due to its compositional structure.
>
> > Q3: In Figure 1, what means the surrounding figures? Maybe we guess it is to visualize the look ahead planning, but I am not sure it is a good visualization for showing that.
>
> We're sorry for the confusion. Yes, it's meant to visualize the look ahead planning. We have updated a better version with fewer frames and more clear annotations to improve the interpretability.
>
> *We sincerely appreciate your constructive comments. Please feel free to let us know if you have further questions.*
>
> Best,
>
> Authors

---

> ### Comment · Reviewer_Dcsa · 2024-11-25
> **Reply to the rebuttal**
>
> First of all, thank authors for preparing concrete rebuttal with additional experiments in the limited time.
>
> Their rebuttals addressed my concerns well, but I have one more question.
>
> - Could you explain why Success Rate@30 is good measurement for checking the performance? It looks showing the efficiency of the plannings.

---

> > ### Author Response · Authors · 2024-11-25
> > **Response to Reviewer Dcsa**
> >
> > Thank you for your prompt feedback and for raising this important question. We are glad to hear that our initial responses addressed your concerns. Below, we explain the rationale behind using **Success Rate@30** as a performance metric:
> >
> > - **Success Rate is horizon-dependent:** It measures task completion relative to the planning horizon $h$. In our original experiments, we set $h = 60$ to ensure that most baseline methods could complete the task. However, as you correctly observed, this approach limits its usefulness in distinguishing strong baselines. When given sufficient steps, even random or suboptimal methods might eventually escape dead loops, diminishing the metric's discriminative power.
> >
> > - **Rationale for $h = 30$:** From our analysis, we observed that all episodes could be completed in approximately 20 steps in ideal conditions. To better reflect efficiency and practicality, we chose a shorter horizon of $h = 1.5 \times 20 = 30$. This adjustment emphasizes whether agents can complete tasks within a reasonable and finite number of steps, making the metric more effective for comparing the performance of strong planning methods, and preventing misleading results caused by noise or outlier behavior.
> >
> > We hope this clarifies the reasoning behind our choice of Success Rate@30. Please let us know if you have further questions or require additional clarification.
> >
> > Best,
> >
> > Authors

---

> > > ### Comment · Reviewer_Dcsa · 2024-11-25
> > > **Reply to the additional comment**
> > >
> > > Thank you authors for quick comments.
> > >
> > > Your explanation makes sense and we agree that it can show the effectiveness of COMBO more clearly.
> > >
> > > We keep our score to lean to the acceptance of this paper.

---

### Official Review · Reviewer_Rt2j · 2024-11-01

**Soundness:** 2
**Presentation:** 3
**Contribution:** 2
**Rating:** 6
**Confidence:** 3

**Summary:**

The paper propose to use compositional video diffusion model as "world model" for an environment with several agents cooperatively solving the task. In addition to learning scores for each agent independently, the diffusion model is trained only on regions reachable by an agent. Such model was used in COMBO agent that is effectively planning by fine-tuning VLM to propose reasonable action given current states while also fine-tuning VLM for intend prediction and for the evaluation of the current state.

**Strengths:**

- The problem of join cooperation of several agents is interesting, and the approach for world modeling by merging observations from different agents seems plausible.

- Decomposition of the full state to regions that can be affected by each agent (while not correct in general e.g. turning on light would change the whole image, however it is a reasonable assumption that the overall scene in effected by agents mostly independently)

**Weaknesses:**

- Scaling loss with the reachability assumes that reachability is provided externally, in real world agents "reachability" should be additionally estimated / discovered from the exploration data.  It would be great if the authors would cover better how to discover the reachability regions if they are not provided. Also, what about regions what are not reachable by any agent? In current formulation, it is not clear if those regions are modeled or not in the world model.

 - Fine-tuning of VLM on the "collected short rollouts" potentially lead to agent that memories how other agents behave. While this approach can work, it would always require fine-tuning "intend prediction" if the agent is changing policy. It would be great if the authors can show that VLM can use some of the initial observations of the agent behaviors to adapt its predictions  on the fly (e.g. with in context learning).

- In both of the TDW-Game and TDW-Cook performance is saturated learning to agents that perform near (and in case of the TDW-Cook somehow better 22.8 vs 24.0) than Oracle Cooperator. Thus, it would be great if authors can increase the difficulty of those tasks and show possible failure cases of their approach. For example, currently world model assumes that visual information and actions is enough to predict the next state. However, for in closer to real world scenarios some parameters are hidden (e.g. objects mass) and thus effective world model would need to deal successfully with estimation of such parameters and using them for the determining of the optimal actions.

**Questions:**

- Please describe in more details on what data intend and outcome evaluation was fine-tune on? How realistic collection of such data for real-world agents?

- Potentially more details could be provided on modification of 2D FetchQ environment, as I didn't find any in the appendix.

- Why Co-Gaild baseline is used in this task but not used on the original environments?

- Is usage or fine-tuning of VLM really necessary for these tasks? If so, does this fine-tuning leads to generalizable action generation /agents intends? For example, how robust agents would follow recipes if some ingredients are not seen in the training? Without studying and showing such generalizations, it is not clear why LLM are needed and if more simple supervised models would do the same job.

- Some formulas could be better connected. E.g in section 4 X is used for world model state, while in 5 s is used.

---

> ### Author Response · Authors · 2024-11-24
> **Response to Reviewer Rt2j [1/3]**
>
> *We sincerely thank the reviewer for the time to read our paper, and we appreciate your positive and constructive comments! We address your questions in detail below and updated our paper according to your suggestions.*
>
> > Q1: Scaling loss with the reachability assumes that reachability is provided externally, in real world agents "reachability" should be additionally estimated/discovered from the exploration data. It would be great if the authors would cover better how to discover the reachability regions if they are not provided.
>
> Thanks for the valuable question!
> - In our specific setting, reachability is defined by the task rules, where agents are restricted to accessing only one side of the table and must collaborate closely to complete the task. Therefore, we utilize the agents' reachable regions to define the coefficient matrix.
> - That said, this is not the only approach to setting the loss coefficient matrix based on reachability. When such information is unavailable, advanced object detection and segmentation models like GroundingDINO [1] and SAM2 [2] can be employed to preprocess the data to label the semantic regions and set the coefficient matrix accordingly.
>
> > Q2: what about regions that are not reachable by any agent? In the current formulation, it is not clear if those regions are modeled or not in the world model.
>
> Thanks for bringing up this issue!
> - In the first stage of training the CWM, agent-dependent loss scaling is employed to better capture $P(X|x_0,a_i)$. As detailed in Appendix A.3, the coefficient matrix $C$ is set to 2 for reachable regions and 1 for other regions. This encourages the model to focus more on relevant pixels while avoiding the complete exclusion of unreachable regions.
> - In the second stage, during compositional training, all regions in the image are treated equally. This includes regions that are unreachable by any agent, which are also explicitly modeled in the CWM.
>
> > Q3: Fine-tuning of VLM on the "collected short rollouts" potentially lead to agent that memories how other agents behave. While this approach can work, it would always require fine-tuning "intend prediction" if the agent is changing policy. It would be great if the authors can show that VLM can use some of the initial observations of the agent behaviors to adapt its predictions on the fly (e.g. with in context learning).
>
> Thank you for the thoughtful feedback! We agree that it is desirable for the intent tracker module to generalize to unseen cooperators and adapt its predictions dynamically based on historical observations.
>
> - Our approach employs **a shared intent tracker module** across all tasks and cooperators. The test-time cooperator's policy is agnostic to our agent during training, meaning the intent tracker **must rely on observation history to adapt its predictions** and track the intents of diverse cooperators effectively. As demonstrated in Table 1, our agent successfully collaborates with cooperators following different policies.
> - To address this point more explicitly, we **add a new qualitative result** in Appendix C.2 with Figure 13 which illustrates how the intent tracker module adapts its predictions based on historical observations. For example, given the same current state, the intent tracker accurately predicts the next action of Agent Bob by leveraging its observation history, which reveals a preference for preparing his recipe first or helping Agent Alice first. With accurate intent tracking, it becomes possible for Agent Alice to select the best action to take. This adaptation showcases the model's ability to generalize and predict dynamically without relying solely on memorizing.
>
> >Q4: Please describe in more details on what data intend and outcome evaluation was fine-tune on? How realistic collection of such data for real-world agents?
>
> Thanks for bringing up this issue! We provide more details below and also added them in Appendix C.1.
> - We generated 4k training environments by randomly sampling a recipe in TDW-Cook and a puzzle box in TDW-Game for each agent. All objects were initialized at random positions on the table. Scripted planners with randomness were then used to play out the episodes and collect the state and action histories.
> - For the Intent Tracker, we collected 40k short rollouts consisting of three images of consecutive observations and a textual description of the next actions of all the agents, converted by a template given the action history.
> - For the Outcome Evaluator, we collected 138k data consisting of one image of the observation and a textual description of the state of each object in the image and the heuristic score, converted by a template given each object's location.
> - This dataset collection pipeline is transferable to real-world robotics tasks. For instance, in practical scenarios, robots can follow a similar approach by utilizing predefined task rules and leveraging planners to simulate and annotate task executions in diverse settings.

---

> > ### Author Response · Authors · 2024-11-24
> > **Response to Reviewer Rt2j [2/3]**
> >
> > >Q5: Is usage or fine-tuning of VLM really necessary for these tasks? If so, does this fine-tuning leads to generalizable action generation /agents intends? For example, how robust agents would follow recipes if some ingredients are not seen in the training? Without studying and showing such generalizations, it is not clear why LLM are needed and if more simple supervised models would do the same job.
> >
> > We appreciate your valuable question! We experimented with alternative implementations of the three planning sub-modules and hope to provide more insights for the community. The results and discussions are also incorporated into **Appendix C.3**
> > - Following your suggestion, we **implemented a simpler supervised model** and trained it using the same dataset we employed for fine-tuning the VLM and evaluated it on the test data. This model encodes the input image using a VIT-B/16 CLIP encoder and the textual task description using the BERT-base-uncased encoder. The encoded outputs are concatenated and passed through a two-layer fully connected network with a hidden dimension of 256.
> >   - For the **action proposer**, the output is a multi-hot vector of 157 dimensions, representing all possible actions.
> >   - For the **intent tracker**, the output is a vector of 4 × 157 dimensions, capturing intents across agents.
> >   - For the **outcome evaluator**, the output is a scalar representing the heuristic score.
> >
> > For each component, we selected the best result across 10 training epochs. The results of each submodule's prediction accuracy, shown in the table below, indicate that **VLMs achieve superior performance compared to this supervised model**. This can be attributed to the common-sense knowledge embedded in the VLM during its pretraining on large-scale internet data, as well as its superior ability to integrate the two modalities effectively.
> >
> > |  | LLaVA-1.5-7B | Simpler Supervised Model |
> > |------|--------|----------|
> > | Action Proposer| **98.3** |  90.3
> > | Intent Tracker| **79.4** | 73.2
> > | Outcome Evaluator| **99.7** | 62.7
> > - The task description is in text format and the observation is in image format, so it's straightforward to use a VLM as the backbone for the submodules. We finetuned LLaVA-1.5-7B **with LoRA for one epoch**.
> > - In our added real-world experiments, we find that the VLM fine-tuned only with simulator collected data is able to generalize to the real images as well, showing a great advantage of using internet-scale data pre-trained VLM, which is also observed in [3].
> >
> > > Q6: In both the TDW-Game and TDW-Cook performance is saturated leading to agents that perform near (and in the case of the TDW-Cook somehow better 22.8 vs 24.0) than Oracle Cooperator.
> >
> > Thanks for pointing this out!
> >
> > - We apologize for any confusion regarding the term "Oracle Cooperator" in the original manuscript. To clarify, this term **does not represent the optimal performance** achievable on the benchmark. To improve clarity, we have renamed this baseline to "Shared Belief Cooperator." This baseline is implemented using a planner that shares the same policy as the cooperator. While it has access to the oracle state of the environment and other agents' policies, **it does not guarantee optimality**. For instance, in the case of TDW-Cook Cooperator 1, which adopts a "selfish" strategy by prioritizing actions related to its own recipe, another cooperator following the same "selfish" policy would not be considered the optimal cooperator.

---

> > > ### Author Response · Authors · 2024-11-24
> > > **Response to Reviewer Rt2j [3/3]**
> > >
> > > > Q7: It would be great if authors can show possible failure cases of their approach. For example, currently world model assumes that visual information and actions is enough to predict the next state. However, for in closer to real world scenarios some parameters are hidden (e.g. objects mass) and thus effective world model would need to deal successfully with estimation of such parameters and using them for the determining of the optimal actions.
> > >
> > > Thanks for this valuable suggestion!
> > >
> > > - As shown in Table 3, the **Composable World Model (CWM) fails in 25% of cases** when predicting the next state conditioned on the actions of four agents. Following your suggestion, we **have analyzed these failure cases in Appendix A.5**, providing two new figures that illustrate common issues such as **Image Quality** and **Misunderstanding Errors**. We hope these insights will guide future improvements.
> > > - We also want to emphasize the inherent difficulty of simulating the effects of multiple sets of actions on the world state using only visual information and actions. Previous methods, as shown in Table 3, have achieved success rates of only **20%–25%** in such settings. Our method significantly improves this performance to **75%–100%** by leveraging composable generation and agent-dependent loss scaling.
> > > - Lastly, we agree that incorporating hidden physical parameters, such as object mass, is an essential direction for making the task closer to real-world scenarios. Extending our framework to include **physics-aware video generation approaches**, such as [4][5], is a promising avenue for future work and could further improve the capability of world models to handle hidden variables effectively.
> > >
> > >
> > > > Q8: Potentially more details could be provided on modification of 2D FetchQ environment, as I didn't find any in the appendix.
> > >
> > > Thanks for the suggestions! We've added a section in Appendix A.1.2 to provide more details on the 2D FetchQ environment. Specifically, we adapted this environment with a visual observation space and high-level action space to evaluate our method.
> > > - Observation Space: $128\times128$ RGB image.
> > > - Action space: *move up*, *move down*, *move left*, *move right*, *wait*.
> > >
> > > > Q9: Why Co-Gail baseline is used in this task but not used on the original environments?
> > >
> > > Thanks for bringing up this issue! Following your suggestion, we **implemented Co-GAIL in our TDW environment**, but it yielded similarly poor performance to another MARL baseline, MAPPO.
> > > - While Co-GAIL is the state-of-the-art (SOTA) method in the original 2D FetchQ environment, it relies on a simpler state representation and requires interacting with the environment online for **6 million steps** to optimize its policy using PPO. While this approach is feasible in simpler environments, it becomes impractical in embodied multi-agent simulations due to the significantly longer time required to sample each step. Specifically, in our TDW environments, the slower sample collection in embodied multi-agent simulations, combined with the larger state representation (e.g., RGB images), poses significant challenges for reinforcement learning methods like Co-GAIL and MAPPO.
> > >
> > > > Q10: Some formulas could be better connected. E.g in section 4 X is used for the world model state, while in 5 s is used.
> > >
> > > Thanks for the suggestion! We've modified the manuscript to use $s$ consistently to represent the world state and explain the connection clearer in Eq (4) and checked all the formulas again.
> > >
> > > [1] Grounding DINO: Marrying DINO with Grounded Pre-Training for Open-Set Object Detection. ECCV24
> > >
> > > [2] SAM 2: Segment Anything in Images and Videos. Meta
> > >
> > > [3] Video Language Planning. ICLR24
> > >
> > > [4] PhysGen: Rigid-Body Physics-Grounded Image-to-Video Generation. ECCV24
> > >
> > > [5] PhysDreamer: Physics-Based Interaction with 3D Objects via Video Generation. ECCV24
> > >
> > >
> > > *We sincerely appreciate your insightful comments and suggestions. Please feel free to let us know if you have further questions.*
> > >
> > >
> > > Best,
> > >
> > > Authors

---

> > > > ### Comment · Reviewer_Rt2j · 2024-11-25
> > > > **Reply to the rebuttal**
> > > >
> > > > I thank authors for their work addressing my questions and concerns. The authors provided additional information in appendix, clarified the paper and provided useful experimental evidence. I appreciate additional failure cases and other supervised models baselines (potentially also add computational requirements both for fine-tuning of LLaVa and training of those models) as I think those could be useful for the potential readers of the paper.
> > > >
> > > > > ...When such information is unavailable, advanced object detection and segmentation models like GroundingDINO [1] and SAM2 [2] can be employed to preprocess the data to label the semantic regions and set the coefficient matrix accordingly.
> > > >
> > > > Thank you for your answer. It is still not clear to me how exactly semantics of the surrounding (e.g. GroundingDINO) is useful for  the reachability. But overall I agree that one can rely on zero-shot VLM to bootstrap from, but I guess it is still important to test if the prediction made by VLM are actually a case (e.g. having an agent that learns its own reachability while interacting with the environment).
> > > >
> > > > > Our approach employs a shared intent tracker module across all tasks and cooperators. The test-time cooperator's policy is agnostic to our agent during training, meaning the intent tracker must rely on observation history to adapt its predictions and track the intents of diverse cooperators effectively. As demonstrated in Table 1, our agent successfully collaborates with cooperators following different policies.
> > > >
> > > > My question was more about, if the agents see another agent with previously unseen behavior (e.g. not one of the two provided in the training data). How would it adapt to this new policy?
> > > >
> > > > While several assumptions are not realistic (such as reachability and the availability of the dataset with all possible agents behaviors), I think they could be talked in the future work. Thus, I will recommend acceptance, while keeping my score.

---

> ### Author Response · Authors · 2024-11-26
> **Response to Reviewer Rt2j**
>
> Thank you for your continued valuable feedback and insightful suggestions to strengthen our work. We are glad you found our responses and updates useful. Based on your suggestions, we have incorporated computational requirements into Appendix C and provide further clarifications below to address your remaining concerns.
>
> > Reachability and Agent-Dependent Loss Scaling
>
> Thank you for raising the point regarding the assumption of reachability. We provide additional clarification and discuss potential approaches:
>
> - **Reachability is not strictly required to set the loss coefficient matrix:** The key idea of this technique is to encourage the model to focus more on the pixels related to the action prompt. There are also other possible ways to leverage this technique. One promising way is to use advanced vision models to identify the action prompt-relevant regions in an image (e.g., "the lettuce" in the action prompt "Bob pick up the lettuce") and adjust the loss scale for those regions.
> - **Alternative approaches:** We agree with your suggestion of discovering reachability during exploration. One promising avenue is bootstrapping from zero-shot vision-language models (VLMs), where an agent could learn reachability by interacting with the environment.
> - **Effectiveness without agent-dependent loss scaling:** Even without this agent-dependent loss scaling, our proposed method still significantly improves the previous methods by leveraging the composable generation. As shown in Table 3, we observe improvements in success rates from **20%–25%** to **55\%-70\%** in the challenging setting of simulating the effects of multiple sets of actions on the world state.
> - We've also incorporated this discussion into Appendix A.3 in the manuscript.
>
> > Adaptability of the Intent Tracker to Unseen Agent Behaviors
>
> We appreciate your clarification on the question about adapting to agents with previously unseen behaviors. Below, we outline how our method addresses this challenge:
>
> - **Intent tracker adaptability**: Our intent tracker leverages observation history to adapt its predictions dynamically, even for cooperator behaviors not encountered during training. While the tracker is trained on a limited set of scripted planners (e.g., the 4-agent TDW-Game setting), it generalizes reasonably well to new behaviors due to its reliance on cooperator-independent observation patterns.
> - **Generalization across settings**: As demonstrated in Table 5, the same intent tracker was applied without retraining in 3-agent and 2-agent versions of TDW-Game, where cooperators followed policies unseen during the training phase. The intent tracker module demonstrated adaptability, achieving correct intent prediction rates of **75.8\%** in the 2-agent cooperation scenario and **64.6\%** in the 3-agent cooperation scenario. Among the wrong predictions, 20\% attributes to the intent prediction of "wait" while it shouldn't be. These results underscore the intent tracker module’s ability to adjust to new behaviors.
> - **Future improvements:** We acknowledge that further generalization to more diverse and complex cooperator behaviors remains a challenge and is a promising avenue for future research. Enhancements such as Population-Based Training [1] and Behavior Diversity Play [2] could increase the diversity of cooperators during training, thus improving robustness. Additionally, incorporating more sophisticated modeling techniques, such as the Bayesian Theory of Mind [3], could enable better inference and adaptability in multi-agent collaborations.
> - We've also incorporated this new result and discussion into Appendix C.2 in the manuscript.
>
> [1] Human-level performance in 3D multiplayer games with population-based reinforcement learning. Science,2019
>
> [2] Adaptive Coordination in Social Embodied Rearrangement. ICML,2023
>
> [3] Too Many Cooks: Bayesian Inference for Coordinating Multi-Agent Collaboration. Topics in Cognitive Science,2021
>
> *We sincerely appreciate your insightful comments and detailed suggestions. Please feel free to let us know if you have further questions or require additional clarification.*
>
> Best,
>
> Authors

---

### Official Review · Reviewer_yTaz · 2024-11-04

**Soundness:** 3
**Presentation:** 3
**Contribution:** 3
**Rating:** 8
**Confidence:** 3

**Summary:**

This paper proposes a compositional world model for multi-agent cooperation, leveraging large generative models and compositional diffusion models to build accurate simulations with an arbitrary number of agents.

The authors present a novel framework called COMBO, which involves the following procedures:

1. **Estimate the Current World State**: Use a diffusion model to infer the state from partial egocentric observations.
2. **Action Proposal and Evaluation**: Employ a pretrained Vision-Language Model (VLM) to suggest actions, predict other agents’ intentions.
3. **Future Frame Generation**: Produce future frames based on the current world state image and a text prompt, representing joint actions of multiple agents. This is performed by the Compositional World Model, leveraging a compositional diffusion architecture.
4. **Evaluate and Plan**: Assess the predicted outcomes from the compositional world model using the VLM, and plan with a tree search algorithm based on these evaluations.

To assess their model, the authors test their approach on three datasets—TDW-Game, TDW-Cook, and 2D-FetchQ—and compare it to several baselines, including a VAE-based world model, MAPPO, CoELA, and LLaVA. The results demonstrate that the COMBO framework significantly outperforms these baselines, especially in planning capabilities.

**Strengths:**

**Novelty**. Suggested framework, named COMBO, offers a unique solution to the multi-agent planning problem by utilizing compositional world modeling for accurate simulation.

**Clear Framework Explanation.** The framework is presented clearly and is easy to follow. Each setting and procedure is understandable through Figure 3 and Algorithm 1. The roles of each module are well-explained in the text and formulation.

**Well-designed Experiments And Clarified Implications of Results.** The experimental setup and baseline choices effectively test the COMBO approach's capabilities. The results demonstrate impressive performance and highlight the necessity of the design choices in their architecture.

**Weaknesses:**

**Lack of Figure Clarity and Interpretability.** The figures in the paper are unclear and difficult to interpret. Illustrations should enhance understanding, but these require reading the text to decipher them. For instance, Figure 1-(b) displays a random assortment of images without any labels. I believe this can be resolved by adding explicit labels for sequential processes that each frame means. Similarly, Figure 4 presents consecutive frames without explanations. Adding annotations, e.g. state, instruction, prediction t=1~3, would make the figures more informative.

**Limited Scalability and Impact.** Despite its impressive performance, the proposed model may struggle to scale in more realistic scenarios, such as handling low-level controls with continuous action spaces and operating without access to ground truth environmental labels during training. I believe that those compositional world modeling abilities seem reliant on structured inputs through prompts, which aren't typically available in realistic multi-agent cooperation setups.

**Questions:**

- Does the action prompt to CWM use a cropped image token along with text, as depicted in Figure 4, or is that just an illustration?
- As mentioned in the weaknesses, I have concerns about potential scalability. Can this be generalized to continuous action spaces, such as in robotics tasks? If so, it would be beneficial to include experiments on this. Additionally, I wonder if the CWM training setup will be applicable in realistic scenarios. In my opinion, we might only have access to egocentric observations and action history during training.

---

> ### Author Response · Authors · 2024-11-24
> **Response to Reviewer yTaz**
>
> *We appreciate the positive and constructive comments from you! We address your questions in detail below and have updated our paper according to your suggestions.*
>
> > Q1: Lack of Figure Clarity and Interpretability.
>
> Thanks for the detailed and actionable suggestions to improve our work. We've reorganized Figure 1-(b) with fewer images and added explicit labels, and Figure 4 with added annotations to improve the interpretability of the figures.
>
> > Q2: Despite its impressive performance, the proposed model may struggle to scale in more realistic scenarios, such as handling low-level controls with continuous action spaces.
>
> Thanks for bringing up this issue!
> - Our compositional framework is designed to improve multi-agent task planning capabilities and can easily integrate various low-level control policies, including reinforcement learning and trajectory optimization for robotics tasks.
> - We **add a new real-world experiment** following your suggestion, and the results show our method is **capable in real-world scenarios** involving low-level control and non-trivial generalization. Two figures showing our task setup and results and more details can be found in Appendix B.  We also upload the demo video in the updated supplementary materials.
>     - We instantiate the TDW-Game benchmark in the real world with two agents. A human and an XArm sit on the upper and right sides of the table, each can only reach the objects on his side of the table and must cooperate to pass and place 3-5 puzzle pieces scattered randomly on the table into the correct puzzle box according to visual clues such as the shape. In each step, the human or the robot can only pick or place one object within his reachable region. With 5 trials of experiments, our method succeeded in 4 trials with an average step of 8.4.
>
> > Q3: I wonder if the CWM training setup will be applicable in realistic scenarios. In my opinion, we might only have access to egocentric observations and action history during training. I believe that those compositional world modeling abilities seem reliant on structured inputs through prompts, which aren't typically available in realistic multi-agent cooperation setups.
>
> Thanks for the valuable question.
> - Our environments are built on top of ThreeDWrold[1], which is a physically realistic simulation platform. Therefore our training pipeline could also be employed in the real world.
> - The only data we need to collect to train the CWM are ego-centric observations, action history, and orthographic views, which could all be collected in the real world with RGBD cameras.
> - The input to CWM is just converting the actions of agents into text through a template, which is easy to get in even real world.
>
> > Q4: Does the action prompt to CWM use a cropped image token along with text, as depicted in Figure 4, or is that just an illustration?
>
> We're sorry for the confusion.
>
> - The cropped image in Figure 4 is just an illustration for better interpretability.
> - The action prompt to CWM is pure text describing the joint actions, such as "Alice pick up the lettuce. Bob pick up the pickle slice."
>
> [1] ThreeDWorld: A platform for interactive multi-modal physical simulation. NeurIPS21
>
> *We sincerely appreciate your detailed and constructive comments. Please feel free to let us know if you have further questions.*
>
> Best,
>
> Authors

---

> > ### Comment · Reviewer_yTaz · 2024-11-25
> >
> > I thank the authors for their thorough response and insightful additional experiments. The detailed explanations have addressed my main concerns regarding Questions 1, 3, and 4.
> >
> > The primary concern raised in Q2 was the applicability of low-level cooperation in continuous action spaces, such as multi-agent collision avoidance problems. Although the author demonstrated this approach in real-world scenarios, the concern remains relevant.
> >
> > Nevertheless, although those limitations remain, the additional experiments demonstrate the COMBO framework’s potential to handle a non-trivial range of real-world scenarios by leveraging it as a high-level policy integrated with off-the-shelf low-level controllers.
> >
> > Overall, I believe this paper is worthy of acceptance, and I will update my score accordingly.

---

> > > ### Author Response · Authors · 2024-11-26
> > > **Response to Reviewer yTaz**
> > >
> > > Thank you for your thoughtful feedback and for recognizing our efforts in addressing your concerns. We appreciate your acknowledgment of the COMBO framework’s potential in real-world scenarios.
> > >
> > > Regarding the applicability to continuous action spaces, we agree this is an important area for future exploration. While our current work focuses on high-level policy planning, we agree that extending its applicability to low-level cooperation in continuous action spaces, such as multi-agent collision avoidance, would further broaden its utility. We are grateful for this valuable insight and will consider it as a key avenue for future research and development.
> > >
> > > We are grateful for your encouraging evaluation and support for acceptance, and we appreciate your constructive insights.

---

### Author Response · Authors · 2024-11-24
**General Response to All Reviewers and ACs [1/2]**

We thank all the reviewers and ACs for their time and effort in reviewing our paper and giving insightful comments. We're glad to find that **all reviewers share a positive assessment** of our work, and acknowledge our following contributions:

- **A novel framework leveraging compositional world modeling for multi-agent planning problem**

    - offers a unique solution to the multi-agent planning problem by utilizing compositional world modeling for accurate simulation. [yTaz]
    - The problem of joint cooperation of several agents is interesting, and the approach for world modeling by merging observations from different agents seems plausible. [Rt2j]
    - This model stands out from other single-agent, model-based reinforcement learning approaches by explicitly accounting for multi-agent cooperation dynamics [Dcsa]
- **Well-designed Experiments and in-depth analysis**
    - The results demonstrate impressive performance and highlight the necessity of the design choices in their architecture. The results demonstrate that the COMBO framework significantly outperforms these baselines, especially in planning capabilities. [yTaz]
    - Their approach demonstrated superior performance in embodied multi-agent cooperation tasks and, due to the compositional nature of their model, exhibited strong generalization capabilities. [Dcsa]

**Revision Summary**

- We **add a new real-world experiment** following reviewer **yTaz**'s suggestion, and the results show our method is **capable in real-world scenarios** involving low-level control and non-trivial generalization. Two figures showing our task setup and results and more details can be found in **Appendix B**.  We also upload the demo video in the updated **supplementary materials**.
    - We instantiate the TDW-Game benchmark in the real world with two agents. A human and an XArm sit on the upper and right sides of the table, each can only reach the objects on his side of the table and must cooperate to pass and place 3-5 puzzle pieces scattered randomly on the table into the correct puzzle box according to visual clues such as the shape. In each step, the human or the robot can only pick or place one object within his reachable region. With 5 trials of experiments, our method succeeded in 4 trials with an average step of 8.4.
- We **renamed the "Oracle Cooperator" baseline to "Shared Belief Cooperator"** for greater accuracy, as this baseline is implemented using a planner that shares the same policy as the cooperator. While this baseline has access to the oracle state of the environment and the policies of other agents, **it does not guarantee an optimal policy**. For instance, in the case of TDW-Cook Cooperator 1, which adopts a "selfish" strategy by prioritizing actions related to its own recipe, another cooperator following the same "selfish" policy would not be considered the optimal cooperator.
- As pointed out by the reviewer **Dcsa**, the current success rate metric does not provide enough detailed information for comparing the baselines. To address this, we **have introduced an additional metric, success rate@30**, which evaluates the success rate with a shorter planning horizon of $h=30$. The results, shown in the table below, highlight that our method **COMBO clearly outperforms other baselines, including LLaVA**, achieving an average success rate of **0.98** compared to **0.5** for the LLaVA baseline. We've also incorporated this metric in **Table 1** of the manuscript.

|                           | TDW-Game Cooperator 1 | TDW-Game Cooperator 2 | TDW-Cook Cooperator 1 | TDW-Cook Cooperator 2 |
|---------------------------|------------------------|------------------------|------------------------|------------------------|
| **CoELA \***              | 0.90                  | 0.60                  | 0.15                  | 0.05                  |
| **LLaVA**                 | 0.55                  | 0.60                  | 0.35                  | 0.50                  |
| **COMBO (w/o IT)**        | 0.65                  | 0.60                  | 0.80                  | 0.80                  |
| **COMBO (Ours)**          | **1.00**                  | **1.00**                  | **0.90**                  | **1.00**                  |
| **Shared Belief Cooperator \***  | 1.00                  | 1.00                  | 0.95                  | 1.00                  |

---

> ### Author Response · Authors · 2024-11-24
> **General Response to All Reviewers and ACs [2/2]**
>
> - Following reviewer **Dcsa**'s suggestion, we **additionally tested our method and the LLaVA baseline on the 2-agent version of TDW-Game without further training**. The success rate@30 and average steps when cooperating with two different cooperators over 10 episodes are reported in the table below. The results demonstrate that our method **COMBO clearly outperforms LLaVA across all settings of different numbers of agents**, completing all tasks within 30 steps without requiring additional training. These results have also been incorporated into **Table 5** of the manuscript.
>
> |Success Rate@30, Steps | 2-agent | 3-agent | 4-agent |
> | -- | --- | --- |---|
> LLaVA | 0.8, 20.7 | 0.65, 27.4 | 0.58, 28.7
> COMBO | **1.0, 10.5** | **1.0, 15.0** | **1.0, 17.5**
>
> - We **add a new qualitative result** in **Appendix C.2** with a **Figure.13** which illustrates how the intent tracker module adapts its predictions based on historical observations as Reviewer **Rt2j** suggested.
> - We add an additional section in **Appendix C.3** to discuss the alternative implementation choices for the planning submodules and conduct experiments to show **VLMs not only offer better generalizability but also achieve superior performance compared to a simpler supervised model** as Reviewer **Rt2j** suggested. This is also reported in **Table 7** in the manuscript.
>
> |  | LLaVA-1.5-7B | Simpler Supervised Model |
> |------|--------|----------|
> | Action Proposer| **98.3** |  90.3
> | Intent Tracker| **79.4** | 73.2
> | Outcome Evaluator| **99.7** | 62.7
>
> - We add **failure case analysis** in **Appendix A.5**, providing Figure 9 and Figure 10 to illustrate common errors such as *Image Quality* and *Misunderstanding* and discuss the future extension with physics-aware video generation approaches as Reviewer **Rt2j** suggested.
> - We update the **Figure 1(b)** and **Figure 4** for better interpretability as Reviewer **yTaz** and **Dcsa** suggested.
> - We add more details about the 2d-fetchQ challenge in **Appendix A.1.2** as Reviewer **Rt2j** suggested.
> - We add the details of training data collection for the planning submodules in **Appendix C.1** as Reviewer **Rt2j** suggested.
>
> We hope our detailed responses below convincingly address all reviewers’ questions.

---

### Meta-Review · Area_Chair_TfKz · 2024-12-22

**Metareview:**

COMBO presents a significant contribution to embodied multi-agent cooperation through its novel compositional world model approach, demonstrating superior performance across multiple benchmarks. While reviewers initially raised concerns about scalability and generalization capabilities, the authors provided comprehensive responses including new real-world human-robot experiments, quantitative results showing strong generalization to different numbers of agents, and thorough ablation studies demonstrating the necessity of their design choices. The extensive validation across both simulation and real-world scenarios, combined with clear practical impact for multi-agent robotics, makes this a valuable contribution worthy of publication. I recommend accepting this paper.

**Additional Comments On Reviewer Discussion:**

Reviewers raised concerns about figure clarity, real-world scalability, the necessity of VLMs, and generalization capabilities. The authors addressed these through new human-robot experiments showing higher success rate, improved visualizations, ablation studies comparing VLMs with supervised models, and demonstrating generalization across different number of agents. The introduction of success rate@30 metric showed COMBO's clear superiority. Reviewers were satisfied with the responses, maintaining their positive scores, and given the comprehensive validation, particularly in real-world scenarios, the paper warrants acceptance.

---

### Decision · Program_Chairs · 2025-01-22

Accept (Poster)